# Analyzing the performance of metaheuristic algorithms in speed control of brushless DC motor: Implementation and statistical comparison

**Fizza Shafique**[1☉], **Muhammad Salman Fakhar**[1☉], **Akhtar Rasool**[2‡*], **Syed Abdul Rahman Kashif**[1‡]

1 Department of Electrical Engineering, University of Engineering and Technology, Lahore, Punjab, Pakistan,
2 Department of Electrical Engineering, University of Botswana, Gaborone, Botswana

☉ These authors contributed equally to this work.
‡ AR and SARK also contributed to this work.
* rasoola@ub.ac.bw, akhtar@sabanciuniv.edu

**Data Availability Statement:** All relevant data are within the manuscript.

**Funding:** The author(s) received no specific funding for this work.

## Abstract

A brushless DC (BLDC) motor is likewise called an electrically commutated motor; because of its long help life, high productivity, smaller size, and higher power output, it has numerous modern applications. These motors require precise rotor orientation for longevity, as they utilize a magnet at the shaft end, detected by sensors to maintain speed control for stability. In modern apparatuses, the corresponding, primary, and subsidiary (proportional-integral) regulator is broadly utilized in controlling the speed of modern machines; however, an ideal and effective controlling strategy is constantly invited. BLDC motor is a complex system having nonlinearity in its dynamic responses which makes primary controllers in efficient. Therefore, this paper implements metaheuristic optimization techniques such as Whale Optimization Algorithm (WOA), Particle Swarm Optimization (PSO), Ant Colony Optimization (ACO), Accelerated Particle Swarm Optimization (APSO), Levy Flight Trajectory-Based Whale Optimization Algorithm (LFWOA); moreover, a chaotic map and weight factor are also being applied to modify LFWOA (i.e., CMLFWOA) for optimizing the PI controller to control the speed of BLDC motor. Model of the brushless DC motor using a sensorless control strategy incorporated metaheuristic algorithms is simulated on MATLAB (Matrix Laboratory)/Simulink. The Integral Square Error (ISE) criteria is used to determine the efficiency of the algorithms-based controller. In the latter part of this article after implementing these mentioned techniques a comparative analysis of their results is presented through statistical tests using SPSS (Statistical Package for Social Sciences) software. The results of statistical and analytical tests show the significant supremacy of WOA on others.

**Competing interests:** The authors have declared that no competing interests exist.

**Abbreviations:** ACO, Ant Colony Optimization; ANOVA, Analysis of Variance; APSO, Accelerated Particle Swarm Optimization; BLDC, Motor Brushless DC Motor; BPM, Motor Brushless Permanent Magnet Motor; CEC, Congress on Evolutionary Computation; CLT, Central Limit Theorem; CMLFWOA, Chaotic Map Incorporated with LFWOA; CODE, Chaotic Online Differential Evolution; CSTHTS, Cascaded Short Term Hydro-Thermal Scheduling; df, Degree of Freedom; EMF, Electromotive Force; FOPID, Fractional-Order Proportional Integral Derivative; GA, Genetic Algorithm; HSD, Honestly Significant Difference; IAE, Integral Absolute Error; ISE, Integral Square Error; ITSE, Integral Time Square Error; LFD, Levy Flight Distribution; LFWOA, Levy Flight Trajectory Based Whale Optimization Algorithm; MATLAB, Matrix Laboratory; MZN, Modified Ziegler-Nichols; NP, Non-Polynomial; PI, Proportional-Integral; PID, Proportional-Integral-Derivative; PSO, Particle Swarm Optimization; SO, Snake Optimizer; SPSS, Statistical Package for Social Sciences; TL, Tyreus-Luyben; WOA, Whale Optimization Algorithm; ZN, Ziegler-Nichols.

# 1 Introduction

## 1.1 Motivation and incitement

The purpose of an electric motor is to perform mechanical work or function by utilizing electrical energy. The invention of the motor manifested at the beginning of the Industrial Revolution [1, 2]. One of the motor types that has gained more popularity is BLDC, also known as permanent magnet DC. The rotor of the BLDC motor comprises permanent magnets and rotates around a fixed armature instead of the moving armature. Because of its construction, information about the rotor position is required for trapezoidal back-EMF [3]. There are two approaches for determining the position of the rotor; the first is the direct detection method, which uses sensors for detection purposes. These sensors should be accurately placed at their required positions because if the sensors are not set accurately, it may cause an increase in the phase current and a reduction of torque [4]. Moreover, the working reliability is also affected by errors in speed estimation because of uneven intervals of commutation [5]. The second one is the sensorless approach, as researchers are moving towards sensorless technology because a sensorless BLDC motor determines the position of the rotor indirectly and the parameters of the speed controller of the motor and its behaviour and characteristics [6].

The problem of speed loss is very common in DC motors. The loss of speed happens during the loading and unloading of the running load. Due to the fluctuation of rate, a controller becomes a basic motor need. A brushless DC motor exhibits a nonlinear dynamic response due to the direct coupling of the rotor's speed with the direct and quadrature axis currents. Hence, the purpose of each control system is to manage, direct, or regulate the system's working in a control loop. Control loop systems are used to remove any disturbance in the system. In contrast, open-loop control systems are incapable of rectifying any disturbance [7]. A control system can be improved by modifying its regulating parameters by observing the system's transient and steady-state response [8]. BLDC motor makes it inefficient to control its speed using a conventional controller because of the nonlinearity in response of the system. High efficiency is the industry's chief demand, so to overcome the effect of noise or disturbances with minimum error, the field of intelligent controllers is selected.

This study's primary goal is to regulate the sensorless BLDC motor's speed. There are many kinds of speed controllers, e.g., PI (proportional-integral) and PID (proportional-integral-derivative) controllers, but there is still some uncertainty because of noise, friction, and other disturbances that may affect the system's response. Therefore, for efficient and optimal response, intelligent controllers are used. Smart controllers are designed using algorithms for the intelligent tuning of conventional controllers like PID and PI [9]. Metaheuristic algorithms are preferred for optimization, especially for solving non-polynomial-hard problems which are also abbreviated as NP-hard [10]. In the past, there were many traditional methods for tuning the gains of PI and PID controllers, such as the Ziegler-Nichols (ZN) method, Modified Ziegler-Nichlos (MZN) method, and Tyreus-Luyben (TL) method [11]. Efficient controllers are highly demanded in the market; therefore, there are many intelligent algorithms-based controllers, e.g., PID controllers are tuned by the help of algorithms, other than these conventional methods for adjusting the gains of the controllers for numerous applications.

## 1.2 Literature review

R. A. Krohling et al. used two genetic algorithms to modify the PID controller tuning for a servo motor. One algorithm is used to minimize the error, i.e., calculated by using ITSE (Integral Time Square Error) criteria. Moreover the second objective is to maximize the rejection constraint of disturbance [12]. K. Vanchinathan et al. used the Bat Algorithm for tuning the

fractional-order proportional integral derivative (FOPID) parameters for a sensorless BLDC motor [13]. In 2007, E. H. E Bayoumi et al. published an article on the robust current and speed controller design for a permanent magnet BLDC motor in which the PSO-based tuning method is superior in performance as compared to the conventional tuning method, i.e., the Ziegler-Nichols tuning method [9]. For brushless DC motor speed control, Kiree et al. suggested designing an optimal proportional-integral-derivative (PID) controller using Particle Swarm Optimization (PSO). Their method involves utilizing back electromotive force (EMF) detection, aiming to increase the efficiency of the control system of the motor [14]. Mohammed A. Ibrahim published an article in which the PID controller for BLDC motor speed regulation is adjusted by the Genetic Algorithm (GA). Calculating the integral squared error (ISE) and integral absolute error criterion (IAE) allows for the selection of the best possible tuning of the GA-based PID controller. Results show that the ISE-based method is more reliable for sensitive applications [15]. Dutta et al. published an article about the implementation of Grey Wolf Optimizer to tune PID controller for BLDC motor speed control [16]. I. Anshory et al. did research to develop an efficient way to improve the BLDC motor's speed control using the Bat algorithm (BA) to tune the PID. This artificial controller is more efficient than conventional controllers [17]. Estrela et al. introduced an updated version of WOA (Whale Optimization Algorithm) to determine the best method for tuning the PID controller for the DC motor's speed with the least settling time. It is evidenced that the proposed strategy is more effective than the canonical form of WOA in controlling the pace of the BLDC motors; moreover, the steady state is also minimal [18].

E. Çelik in 2024 proposed a new approach to tune cascade 1PDf-PI controller for BLDC motor using Snake Optimizer (SO). The study has shown better performance as compared to the traditional PI and 1PDf controllers, especially under varying speed reference and presence of external torque [19]. In 2022, Aoudni Y. did a study to enhance the resiliency of PID controllers in voltage and current control in such specialized domains of application. This study highlights the importance of advanced optimized techniques of controlling strategies for complex photonic and optical applications [20]. In 2022, Rodríguez-Molina et al. proposed a new approach to finely tune the speed controller if BLDC motor. In this approach a chaotic online differential evolution (CODE) algorithm to control the gains' tuning of the controller in a chaotic manner. Results show that the CODE approach is very effective even in dynamic and uncertain conditions [21]. M. A. Iqbal et al. in 2021 investigated that how the results of the Cascaded Short Term Hydro-Thermal Scheduling (CSTHTS) problem are improved by using different 16 variants of Accelerated PSO and one modified version of PSO. The obtained results are better than already existed results in literature. On the basis of results, APSO and its variants perform well as compared to PSO [22].

### 1.3 Contribution

This study is about using nature-inspired metaheuristic algorithms, including PSO, APSO, ACO, WOA, and LFWOA; moreover, LFWOA incorporated with chaotic maps, to improve the efficiency of the speed controller (i.e., PI controller) for a sensorless brushless permanent magnet DC motor. The aim is the error (e) minimization by using ISE criteria, i.e., in Eq (1) and giving optimal values for tuning parameters of the controller to increase its efficiency.

$$\min (f) = \int_1^t e(\tau)^2 d\tau \tag{1}$$

where "$\tau$" is the time interval (from 1 to t) for which the error (the variation between the

output "y(t)" and the referenced value "ref(t)") is defined in Eq (2).

$$e(t) = ref(t) - y(t) \tag{2}$$

Furthermore, this article creates a comprehensive framework for the evaluation of the performance of these algorithms. The analytical comparison is done on the basis of the value of cost function that is minimization of error based on ISE criteria. To check the statistical significance, hypothetical testing is performed using statistical tests like ANOVA (Analysis of Variance), Mann Whitney U test, t-test, Wilcoxon signed-rank test, Friedman test, and Friedman aligned ranks test in SPSS software.

## 1.4 Paper organization

The paper starts off with an Introduction, which describes that how crucial it is to control the speed of the BLDC motor, and what is the primary goal of this study. This sections gives a brief overview of the content of this paper, the background study comprises of similar work did by other researchers around the world, and what is the role of this study in the field of optimizing the speed control of the BLDC motor. Section 2 and section 3 are about the mathematical model of the BLDC motor and PI controller respectively. The section 4; Methodology describes the used techniques in this paper, the procedure of tuning the gains of the controller with the help of metaheuristic algorithms, and after discuss briefly the statistical analysis comparison of the performance of discussed algorithms. Section 5 compromises of the simulated results and tabular data describes results of different statistical tests (T test, Mann Whitney U test, Wilcoxon signed-rank test, Friedman test, One-way ANOVA test), and Friedman aligned ranks test (improved version of Friedman test). In the last section a concluded observation is discussed depending on the findings.

## 2 Mathematical model of BLDC motor

Brushless DC motors are also known as Permanent Magnet DC Synchronous Motors [23]. With their fast speed and great torque, these BLDC or brushless permanent magnet (BPM) motors are more energy-efficient. A sensorless BLDC motor is an advanced electromechanical system designed to operate without position sensors. This technology counts on advanced control techniques to find the position of the rotor and commutation timings based on the back electromotive force signals generated with accuracy within the motor winding during operation. This approach offers advantages such as reduced complexity and potentially increased reliability by eliminating the need for physical position sensors while maintaining precise control over the motor's operation. The following equations represent the BLDC motor's mathematical model.

The equations that govern the armature voltages in a 3-phase BLDC motor [23] are;

$$v_{an} = L_s \frac{di_{an}}{dt} + R_s \cdot i_{an} + e_a \tag{3}$$

$$v_{bn} = L_s \frac{di_{bn}}{dt} + R_s \cdot i_{bn} + e_b \tag{4}$$

$$v_{cn} = L_s \frac{di_{cn}}{dt} + R_s \cdot i_{cn} + e_c \tag{5}$$

In the matrix form,

$$
\begin{bmatrix} v_{an} \\ v_{bn} \\ v_{cn} \end{bmatrix} = R_s \begin{bmatrix} i_{an} \\ i_{bn} \\ i_{cn} \end{bmatrix} + \begin{bmatrix} L_a & L_{ab} & L_{ac} \\ L_{ba} & L_b & L_{bc} \\ L_{ca} & L_{cb} & L_c \end{bmatrix} \frac{d}{dt} \begin{bmatrix} i_{an} \\ i_{bn} \\ i_{cn} \end{bmatrix} + \begin{bmatrix} e_a \\ e_b \\ e_c \end{bmatrix} \tag{6}
$$

where $R = \begin{bmatrix} R_s & 0 & 0 \\ 0 & R_s & 0 \\ 0 & 0 & R_s \end{bmatrix}$.

Let's assume unsaturated windings, minimal iron loss, equal phase winding resistances, constant self-inductance, and equal mutual inductance.

$$
L_a = L_b = L_c = L \tag{7}
$$

$$
L_{ab} = L_{bc} = L_{ca} = M \tag{8}
$$

The above form (6) is reduced to the following matrix form (9);

$$
\begin{bmatrix} v_{an} \\ v_{bn} \\ v_{cn} \end{bmatrix} = R_s \begin{bmatrix} i_{an} \\ i_{bn} \\ i_{cn} \end{bmatrix} + L_s \frac{d}{dt} \begin{bmatrix} i_{an} \\ i_{bn} \\ i_{cn} \end{bmatrix} + \begin{bmatrix} e_a \\ e_b \\ e_c \end{bmatrix} \tag{9}
$$

where $L_s = L - M$. The induced EMFs of BLDC motors are trapezoidal and can be expressed in mathematical expressions such as;

$$
e_a = f_a(\theta_e)\lambda\omega \tag{10}
$$

$$
e_b = f_b(\theta_e)\lambda\omega \tag{11}
$$

$$
e_c = f_c(\theta_e)\lambda\omega \tag{12}
$$

$$
\omega = p\omega_r \tag{13}
$$

In the above equations, "$\omega$" is the angular electric velocity in (rad/sec),"$\omega_r$" is the angular mechanical velocity in (rad/sec), and the number of pole pairs in the motor is denoted by "p".

Eqs (14) and (15) represent the motor torque (T$_e$) relations with back EMFs and speed.

$$
P_e = i_{an}e_a + i_{bn}e_b + i_{cn}e_c \tag{14}
$$

where "P$_e$" is the electric output power.

$$
T_e = \frac{P_e}{\omega_r} \tag{15}
$$

There is a direct relation between the electric torque of the motor and the supplied amount of current, as shown in Eq (16);

$$
T_e = K_e i \tag{16}
$$

where "K$_e$" is the motor torque coefficient, and "i" is the steady state value of current.

The load torque is represented by "$T_L$", the moment of inertia by "J", and the viscosity coefficient by "B" in Eq (17).

$$T_e - T_L = J\frac{d\omega}{dt} + B \cdot \omega \tag{17}$$

Eq (18) displays the BLDC motor's transfer function;

$$T.F = \frac{\omega(s)}{V(s)} = \frac{K_e}{L_s J s^2 + (L_s B + R_s J)s + K_e K_t} \tag{18}$$

where "$K_t$" is the back-EMF constant.

## 3 PI controller

In engineering and automation systems, proportional-integral (PI) controllers are frequently utilized as feedback control mechanisms. Following the receipt of error information, the integral term takes into account the cumulative historical error over time to eliminate the steadystate error and modifies the control signal appropriately. Meanwhile, the proportional term reacts to the current magnitude of the error to provide an immediate corrective action proportionate to the error. This combination offers a balance between responsiveness and stability across various applications, from simple motor speed control to complex industrial processes.

The output of the PI controller can be defined by the equation:

$$y_c t = K_p \cdot e_{ss}(t) + K_i \cdot \int_0^t e_{ss}(\tau)\, d\tau \tag{19}$$

where "$K_p$" is the proportional gain and "$K_i$" is the integral gain of PI, respectively. "$e_{ss}(t)$" is the calculated steady-state error that can be defined as the difference between the desired or reference speed "$\omega_{ref}(t)$" and the output speed "$\omega_{out}$" of the motor in Eq (16):

$$e_{ss}(t) = \omega_{ref}(t) - \omega_{out}(t) \tag{20}$$

## 4 Methodology (Metaheuristic algorithms)

Metaheuristic algorithms are being chosen for designing intelligent controllers. These algorithms are helpful in optimizing the tuning of the parameters of commonly used controllers. Many approaches have been used, but here are canonical versions of APSO, PSO, WOA, and ACO; moreover, the hybrid model of WOA and Levy flight trajectory and chaotic maps used in Levy flight trajectory-based WOA are discussed.

These metaheuristic techniques are preferable on other conventional techniques of tuning the PI controller's gains. Metaheuristic techniques follow an automated process of optimization which makes them different from other heuristic techniques. PSO technique depends on the population of searching agent and with iterative approach these agents move towards the best optimal solution. PSO is prefer because of its simplicity and fast rate of convergence. APSO offers an accelerated approach to converge at possible optimal solution as it has single update equation. ACO is another nature inspired technique which use the foraging behaviour and communication strategy of ants to traverse the whole search space by coordinating the following searcing particles to move towards the best-found solution. WOA offers a balance between exploration and exploitation which helps the search agents to give optimal solution within the search space and prevent premature convergence at any local optima. WOA is highly adaptive for complex and nonlinear problems. LFWOA incorporated Levy flight

behaviour in the canonical version of WOA and enhance the exploration ability of the algorithm by taking random small steps with occasional long steps to avoid convergence at any local optima. Levy flight concepts is helpful in finding optimal solutions to complex and dynamic problems. Moreover, using chaotic map offers a controlled randomization which is helpful in finding optimal solution. The details of each algorithm are discussed in the following section.

## 4.1 Particle Swarm Optimization (PSO) algorithm

Particle Swarm Optimization (PSO) is a technique that draws inspiration from the communal behaviour of fish schools and flocks of birds. Search agents (particles) traverse the search space to locate the best solution. Each particle modifies its position and velocity in response to information from both its neighbours and itself.

Mathematically, the movement of each particle "k" for iteration "t" can be defined in Eqs (21) and (22).

$$v_k(t+1) = w_{\text{inertia}} \cdot v_k(t) + a_1 \cdot r_1 \odot [g_{\text{best}} - s_k(t)] + a_2 \cdot r_2 \odot [s_{k\_\text{local}} - s_k(t)] \qquad (21)$$

In Eq (21), "V=$(v_1,v_2,v_3,\ldots,v_n)$" represents a vector of velocities of each particle, and similarly "S=$(s_1,s_2,s_3,\ldots,s_n)$" a position vector of each particle "$a_1$" and "$a_2$" are the acceleration coefficients, "$r_1$" and "$r_2$" use for randomization $\in [0, 1]$. The position of each particle is updated by using the Eq (22):

$$s_k(t+1) = s_k(t) + v_k(t+1) \qquad (22)$$

In Eq (21), "$w_{\text{inertia}}$" balances the global and local exploration capabilities of the particles. The higher value of $w_{\text{inertia}}$ (i.e., $\in(0, 1)$) helps global exploration, while lower values promote local exploitation, focusing on promising regions. The acceleration coefficients "$a_1$" and "$a_2$" control the particle's own best position "$g_{\text{best}}$" and the neighbourhood's best position "$s_k\_local$", typically the value of "$a_1$" and "$a_2$" is approximately equal to 2. These coefficients guide the particle's movement toward promising solutions while avoiding premature convergence. The update process involves adjusting the particle's velocity based on its previous velocity, the difference between its current position and its personal best, and the difference between its current position and the neighbourhood's best-known position. This balance between exploration and exploitation enables PSO to efficiently navigate the search space, converging towards optimal solutions.

## 4.2 Accelerated Particle Swarm Optimization (APSO) algorithm

Accelerated Particle Swarm Optimization (APSO) presents an advancement of the traditional Particle Swarm Optimization (PSO) algorithm. APSO aims to enhance the convergence speed and exploration efficiency of the PSO algorithm by introducing adaptations to the velocity update mechanism. APSO refines the velocity update equation, commonly utilized in PSO, as depicted in Eq (23) [10]. This innovative approach dynamically adjusts particle velocities during optimization, allowing for a more adaptive exploration-exploitation trade-off.

$$v_k(t+1) = v_k(t) + a_1(r - 0.5) + a_2(g_{best} - s_k(t)) \qquad (23)$$

where "r" is used for randomization $\in [0, 1]$. After updating the velocity of the particle, Eq (22) is used for position updating.

In APSO to further enhance the convergence, the location update process is transformed into a singular step as expressed mathematically in Eq (24).

$$s_k(t+1) = (1 - a_2)s_k(t) + a_2 g_{best} + a_1(r - 0.5) \tag{24}$$

In Eq (24), "r" is a random variable $\in [0, 1]$, and the values of accelerated constants lie approximately in the range from 0.1 to 0.4 for "$a_1$" and 0.1 to 0.7 for "$a_2$" [10].

The formulation of Accelerated Particle Swarm Optimization (APSO) significantly enhances the capabilities of traditional PSO, showcasing its potential for effectively solving complex multidimensional optimization problems in various domains.

## 4.3 Ant colony optimization (ACO) algorithm

ACO (ant colony optimization) metaheuristic technique is also inspired by nature, which is the foraging behaviour of ants to solve complex optimization problems. ACO procedure is based on the collective behaviour of ants as they search for the shortest path between their colony and prey, leaving and following pheromone trails [10, 20]. The mathematical model of ACO involves various steps that are discussed below:

In ACO, there are two issues: firstly, how to decide which route should be opted for, and second, the rate of evaporation of pheromones. The probability "p" of ants at a node decides to select which route, as shown in Eq (25) [10].

$$p_{xy} = \frac{\emptyset_{xy}^{\alpha} \cdot d_{xy}^{\beta}}{\sum_{x,y=1}^{n} \emptyset_{xy}^{\alpha} \cdot d_{xy}^{\beta}} \tag{25}$$

where "$\alpha$" and "$\beta$" are positive parameters, whose values are approximately equal to 2, usually "$d_{xy}$" is the possibility of selecting the same route, and "$\emptyset_{xy}$" represents the concentration of pheromone between node "x" and "y". The concentration of pheromone varies exponentially using the decaying evaporation "$\gamma$" (whose value is in the range of $[0, 1]$) as represented in Eq (26) [10].

$$\emptyset(t) = \emptyset_i e^{-\gamma t} \tag{26}$$

where "$\emptyset_i$" is the concentration of pheromone when t = 0.

In this basic strategy, only the pheromone deposit of the ant is selected, which gives the best solution, and so far, it is considered the most promising region. This strategy shows the reason behind the use of the ACO algorithm for solving complex problems of optimization.

## 4.4 Whale optimization algorithm (WOA)

WOA (whale optimization algorithm) is a metaheuristic searching technique motivated by the foraging action of "humpback whales". The agent-searching method of the WOA is different from other algorithms. The search method of the WOA is termed as the bubble-net attacking method, in which search agents usually hunt by making circular trajectories in spiral shapes or 9-shaped paths. The hunting method of humpback whales has three stages, as mentioned below [24, 25];

- Encircling the prey

- Exploitation phase

- Exploration phase

The mathematical model of these three stages is depicted in Eqs (27)–(34).

In WOA, the first task is to identify the location of the prey as its location is recognized, and then the search agents encircle the prey following the so-far found best or optimum solution as represented in Eqs (27) and (28).

$$\vec{D} = |\vec{C} \cdot \vec{X}_{\text{best}} - \vec{X}(t)| \tag{27}$$

$$\vec{X}(t+1) = \vec{X}_{\text{best}} - \vec{A} - \vec{D} \tag{28}$$

The current iteration number is represented by "t", and "$\vec{X}_{\text{best}}$" represents the position vector of the searching particle that obtained the optimum solution so far, which will update its position iteratively if a better solution is obtained.

"$\vec{C}$" and "$\vec{A}$" are the coefficient vectors that influence the exploitation and exploration of the algorithm and are defined in Eqs (29) and (30):

$$\vec{C} = 2 \cdot \vec{r} \tag{29}$$

$$\vec{A} = 2\vec{a} \cdot \vec{r} - \vec{a} \tag{30}$$

The value of the variable "$a$" is linearly decreasing from the value of 2 to the value of 0, and "r" is the random vector having a value in the range of [0, 1] [24].

In the exploitation phase, search agents have the option to use a feasible one from the available two approaches: the spiral position updating mechanism and the shrinking encircle mechanism [24, 25]. Eq (31) defines the spiral mechanism; the search agents follow this mechanism if the value of $p \leq 0.5$ while uses Eq (28) to adjust search agents' positions if the value of $p \geq 0.5$ where "p" is the probability, i.e., [0, 1].

$$\vec{X}(t+1) = \vec{D}' \cdot e^{bl} \cdot cos(2\pi l) + \vec{X}_{\text{best}} \tag{31}$$

where "l" is the random variable that lies in the range of [-1,1], and the logarithmic spiral shape of the path is defined by the constant "b" [25]. The vector "$\vec{D}'$" represents the distance vector between the search agent and the best solution found so far. $\vec{D}'$ is defined in Eq (32):

$$\vec{D}' = |\vec{X}_{\text{best}} - \vec{X}(t)| \tag{32}$$

Eq (30) represents that $\vec{A}$ decreases in the same pattern as $a$, so at the exploration stage, the value of $\vec{A}$ is ($> 1$ and $< -1$) to compel search agents to move apart from the best search agents to search the whole region of the solution to get the most optimal global solution. If $|\vec{A} > 1|$ so, use Eq (34) while using Eq (28) to update the positions of search agents.

$$\vec{D} = |\vec{C} \cdot \vec{X}_{random} - \vec{X}| \tag{33}$$

$$\vec{X}(t+1) = \vec{X}_{random} - \vec{A} - \vec{D} \tag{34}$$

where "$\vec{X}_{random}$" a search agent is randomly chosen from the population to explore the whole search space globally to avoid convergence at a local optimum.

## 4.5 Levy flight trajectory-based WOA

The Levy flight idea was proposed by the French mathematician Paul in 1937. Levy flight has generalized beyond Brownian motion, the traditional one. Another metaheuristic algorithm named Levy Flight Distribution (LFD) is established by inspiration from the Levy flight

random walk. In the Levy flight algorithm, in the beginning, the best-known location will be set as a starting point, and then the whole new generation will be generated randomly, conferring to the Levy flight. After generating the new generation, this will be evaluated to find the best one [26].

The WOA is modified by incorporating the concept of Levy flight, a new version of the conventional WOA. It prevents premature convergence of the algorithm with a fast and robust approach and enhances the exploration by increasing the search region of the searching particles by increasing the step length. In LFWOA, the search agent's position is updated by following the trajectory used in the Levy flight technique, as shown in Eq (35) [27].

$$\vec{X}(t+1) = \vec{X}_{\text{best}} + \mu sign[c - 0.5] \oplus Levy \tag{35}$$

where "$\mu$" is a uniformly distributed random number and "c", i.e., a variable $\in [0, 1]$, is used for randomization, and the symbol $\oplus$ is used for the entry-wise multiplication. In this above equation, the "sign[c-0.5]" function can give only three values: 1, 0, and -1.

## 4.6 Levy flight trajectory-based WOA with chaotic maps

The WOA promises convergence speed but is still inefficient in finding the global optimum solution. This approach is known as a chaotic strategy or a greedy strategy. With the help of chaotic maps, the chaotic behaviour in algorithms is correlated with parameters using mapping functions. Moreover, the weight factor is also important because it can control speed in speed-updating equations or formulas. Yintong Li et al., in 2019, introduced a weight factor (w) Eq (36) that helps in position update [28].

$$w = sin\left(\frac{\pi t}{2 \cdot Iteration_{max\_number}} - \frac{\pi}{2}\right) + 1 \tag{36}$$

Now the modified version of Eqs (28) and (31) is defined by the following Eq (37):

$$\vec{X}(t+1) = \begin{cases} w \cdot \vec{X}_{\text{best}} - \vec{A} - \vec{D} & p < 0.5 \\ \vec{D} \cdot e^{bl} \cdot \cos(2\pi l) + w \cdot \vec{X}_{\text{best}} & p > 0.5 \end{cases} \tag{37}$$

In Eq (30), the variable "$a$" is an important variable that controls the algorithm's convergence speed and precision; therefore, it is used to increase or decrease the exploration rate or the exploitation rate of the algorithm. Zhong et al. proposed some nonlinear adjustment strategies using some trigonometric functions and geometrical and logarithmic curves to improve the efficiency of the WOA. On the basis of the results, the cosine function in Eq (38) gives better results [29].

$$a = cos\left(\frac{3\pi t}{Iteration_{max\_number}} - \frac{\pi}{2}\right) + 1 \tag{38}$$

## 4.7 Design of proposed controller

Designing the intelligent proportional-integral (PI) controller for a Brushless DC (BLDC) motor involves integrating the metaheuristic algorithm into the tuning process of the controller's parameters. By implementing this advancement, the controller's gains are optimized dynamically, considering the motor's response and performance criteria. This approach aims

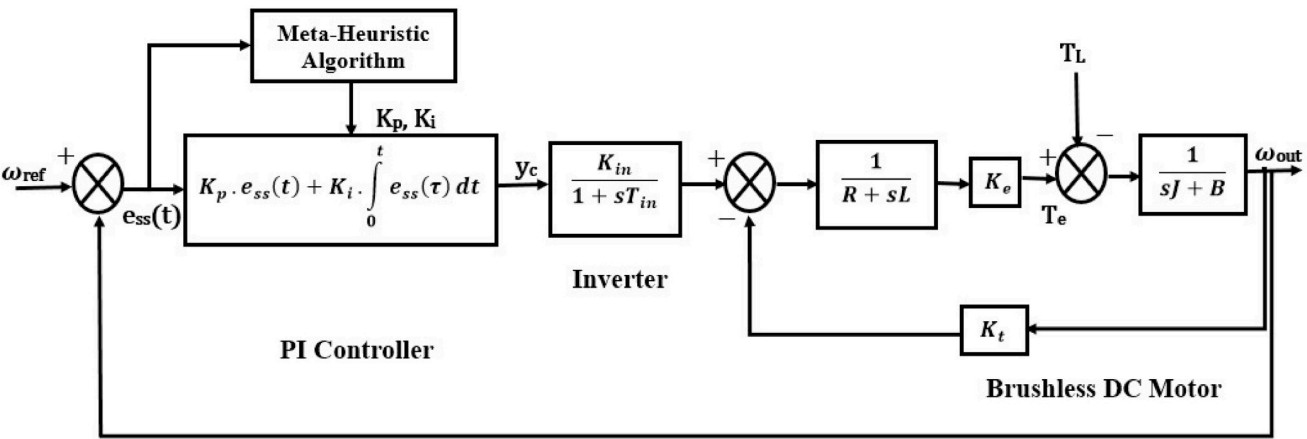

**Fig 1. Proposed PI controller for sensorless BLDC motor.**

to enhance the motor's speed regulation and efficiency by continuously adjusting the PI controller's parameters optimally, ensuring robust and effective control over the BLDC motor's operation. Fig 1, illustrates the structural layout of the suggested controller.

## 4.8 Statistical analysis

In engineering problems, optimization is concerned with finding the best global solution among the different local optima [29]. An optimization problem can be solved by using different metaheuristic algorithms, and their results should then be compared using statistical analysis. In reference [10], the "No Free Lunch theorem" states that there is no surety that the obtained global best solution by one algorithm for one optimization problem is applicable to all problems; therefore, the best one is decided on the basis of statistical analysis to compare the performance of different techniques for a single optimization problem. In IEEE Congress on Evolutionary Computation (CEC) of 2017, 2019, and 2022 some statistical tests are refer to compare different groups of data set [30–32]. In these congresses different problems are referred to evaluate the performance of algorithms in field of optimization. Some of tests referred in CEC 2017, 2019, and 2022 are discussed here briefly. Moreover, some parametric tests are also discussed in the following part because according to Central Limit Theorem (CLT) if the size of data set is greater than 30 then it is assumed that the data is uniformly distributed. In this work, the data set of sized 50 (i.e., the number of time each algorithm is run) is collected for each algorithm. But in this study, this non-parametric test is also performed to give a strong foundation or check to results.

 **4.8.1 T-test.** T-test is a widely used statistical tests which is used to evaluate the significant difference of two groups on the basis of their mean. In early 1900s an English chemist and statistician named William Sealy Gosset published a seminal paper on t-test using the pseudonym "Student". Therefore, the other name of t-test is Student's t-test [33]. T-test is classified as paired samples t-test, independent samples t-test, and one sample t-test. The paired samples t-test is used for the comparison of same data for two different conditions, and the one sample t-test is used to compare a single group against any reference value. The independent samples t-test as name suggests is used to find statistically a significant difference between two non-related groups on the basis of their mean [34]. In this study, the independent samples t-test is applied to compare the means of algorithms in different sets.

**4.8.2 Mann-Whitney U test.** In 1947, professor Henry Mann and his fellow D. Ransom Whitney published their work under the name of "On a test of whether one of two random variables is stochastically greater than the other". In that seminal paper, they proposed a new statistical test named Mann-Whitney U test [35]. This test is basically used for the data set for which there is no need to assume normality in distribution. Mann-Whitney U test finds the significant difference between two groups by comparing their respective mean ranks instead of means.

**4.8.3 Wilcoxon signed-rank test.** The Wilcoxon signed-rank test was developed in 1945 by Frank Wilcoxon [36]. This is another non-parametric test which is used to determine the statistical difference either exist or not between two groups. In this set a positive or negative sign is assigned to each rank. This test can be used for small sized samples.

**4.8.4 One-way analysis of variance (ANOVA).** In 1920s, Sir Ronald A. Fisher introduced a new statistical approach that analyzes the variance among more than one group at a same time [37]. It is a parametric test that is helpful to compare large number of groups. One-way ANOVA provides the information of variance between different groups and variance within each group. One-way ANOVA provides the overall comparison of each group while to separately identify different pairs' comparison Post Hoc tests become necessary. In this study, some these tests are performed like Bonferroni, Scheffe, and Tukey HSD (Honestly Significant Difference) tests.

**4.8.5 Friedman test.** The Friedman test detects the significant difference between more than two groups simultaneously like one-way ANOVA. But the Friedman test is a non-parametric test; therefore, there is no need to hold the assumption of uniformly distributed data. The Friedman test was first introduced by an American statistician, Milton Friedman [38]. The Friedman test is performed on data set which is organized in the form of blocks. The data in each block is ranked first then the ranks across each block are summed. In this test the significant difference level is determine by the comparison of test statistics with chi-square distribution [39]. Due to multiple comparison the risk of type 1 error increases; therefore, post hoc tests (Bonferroni-Dunn, and Holm's Step-Down) are performed manually to provide the information about comparison of individual group.

**4.8.6 Friedman aligned ranks test.** The Friedman aligned ranks test is the modified variant of the conventional Friedman test. In this test, the block effect is neutralized. This effect of block is neutralized by aligning the data and assigning the rank to each group. Next step is the adjustment of data by subtracting the individual rank of each block from the mean rank of whole data. In this test, statistically a significant difference is found by comparing the aligned ranks with the value of chi-square [40]. Like other multi-comparison tests, there are some post hoc tests for the Friedman aligned ranks test. The most commonly used post hoc test is Nemenyi test.

## 4.9 Implementation of metaheuristic algorithms

The following steps describe implementing PSO, APSO, ACO, WOA, and their improved versions to adjust the controller's gain.

I. After configuring the required parameters of the algorithm and defining the boundaries of parameters of the PI controller ($K_p$ and $K_i$) in initialization, the population of search agents is generated using randomization.

II. Define a fitness function based on the main objective of estimating the performance of the metaheuristic technique to tune the PI controller optimally.

III. BLDC motor model runs for each search agent to calculate fitness function.

IV. The main loop of the chosen algorithm generates a solution for each search agent for every iteration and then evaluates the global best find so far for randomly selected search agents.

V. Use the best find solution to update the parameters of the speed controller of the sensorless BLDC motor.

VI. As long as the best solution is achieved, the position of the search agent is updated using the respective position updating equation of algorithms.

VII. The algorithm will be terminated according to stopping criteria (which, in this case, is the limit of iterations as it is an operation research).

The flowchart in Fig 2 shows the implementation of the proposed controller using The Whale Optimization Algorithm. Similarly, the same procedure will be followed for other algorithms. For PSO, use Eq (17) for new velocity and then use this updated velocity to update the position of search agents as shown in Eq (18). For APSO with inertial weight to accelerate the convergence of the algorithm, the single Eq (20) is used to update search agents' position. The same procedure is used for ACO and improved versions of WOA with respective equations like to update the pheromone concentration and use chaotic approaches with levy flight trajectory process to enhance the convergence of the Whale Optimization Algorithm and prevent the solution from sticking to local optima. After implementing the algorithm based approach, next is to do statistical analysis on the gathered data of each algorithm for fixed number of iterations.

### 4.10 Simulation model's specifications

The BLDC motor's mathematical model, discussed above, has been simulated using MATLAB/Simulink software with the values of the parameters mentioned in Table 1 [41].

## 5 Results and discussion

In engineering problems, optimization is concerned with finding the global best solution among different local optima [29]. In optimization, finding feasible solutions that include various local optimum solutions is very hectic work; therefore, different algorithms are used to give the best optimum solution. In MATLAB/Simulink, the sensorless BLDC model is simulated. In this model, suddenly, a load of 0.3 N is applied at 0.1 s; in Figs 3 to 10, the left zoom-in view shows the transient behaviour of the motor, while the right one shows the behaviour of speed curve when a sudden load is applied. The simulated results are discussed below;

### 5.1 Analysis of the results of WOA

The performance of WOA is evaluated for different numbers of search agents for a fixed number of iterations, as displayed in Fig 3.

Table 2 shows that for 50 search agents for $K_p$ = 0.348422343 and $K_i$ = 97.82639568, the value of the cost function is the least, but the overshoot is very high, making the system unstable at up to 0.01451s. After this settling time, the steady state is achieved, and the value of ISE is noted, i.e., 0.283330 (highlighted with bold digits) which is the best choice with a low value of error.

### 5.2 Analysis of the results of LFWOA

The performance of LFWOA is evaluated for different numbers of search agents for a fixed number of iterations, as displayed in Fig 4.

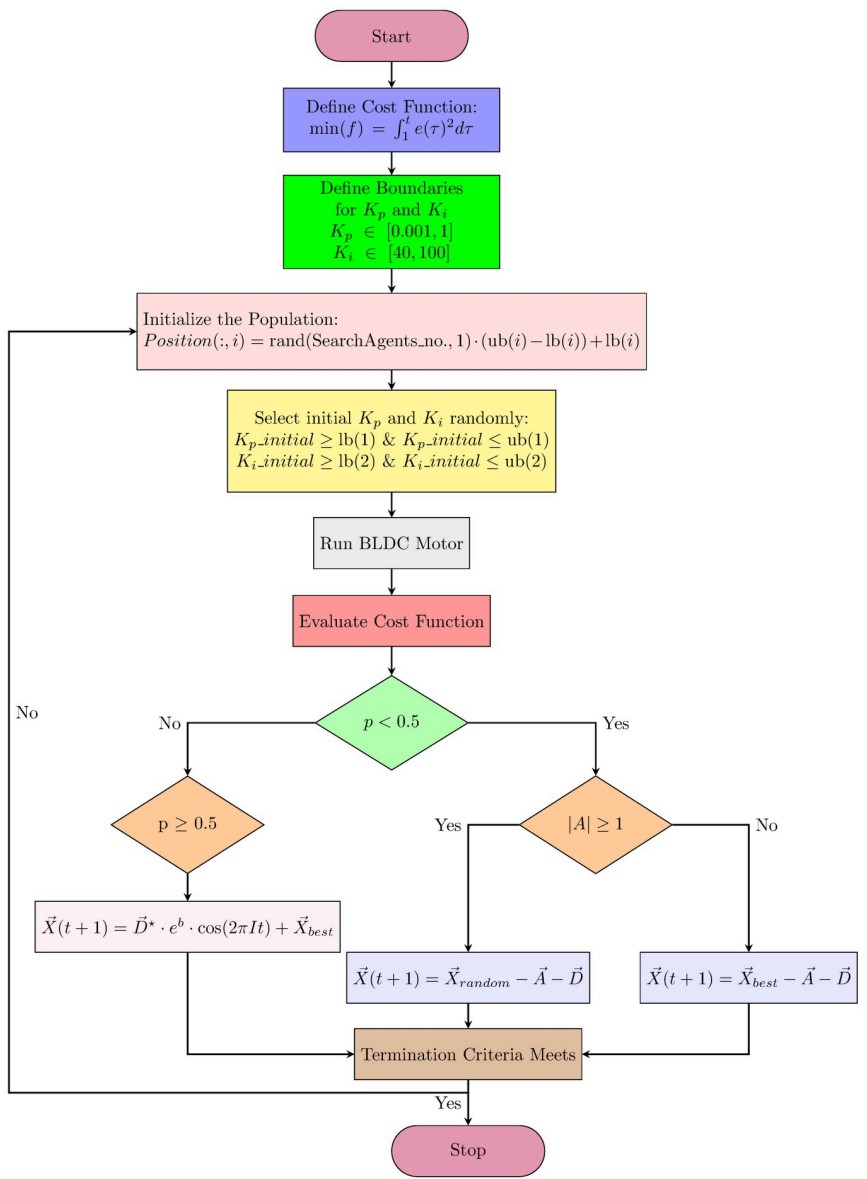

**Fig 2. Flow chart of implementation of metaheuristic algorithm-PI control BLDC motor.**

For 50 search agents in Table 3 for $K_p$ = 1 and $K_i$ = 81.63923175, the value of the cost function is minimal, and against this value, overshoot is very high, which makes the system unstable. Nevertheless, as the settling time is very low, it is the best choice as the system has rapidly achieved its steady state. This optimal solution is shown in Table 3.

## 5.3 Analysis of the results of CMLFWOA

The performance of CMLFWOA is evaluated for different numbers of search agents for a fixed no. of iterations, as displayed in Fig 5.

Table 4 shows that for 50 search agents, the best solution is achieved in which the value of the cost function is minimal but overshoot is very high for these values of parameters of the PI

**Table 1. Sensorless BLDC motor's specifications.**

| Parameters | Values | Units |
|---|---|---|
| Reference Speed | 3000 | rpm |
| Flux Linkage | 0.175 | Vs |
| Torque Constant | 1.4 | Nm/A |
| No. of Poles | 4 | |
| Inertia | $0.8 \times 10^{-3}$ | $kgm^2$ |
| Friction Factor | $1 \times 10^{-3}$ | Nms |
| Phase Resistance of Stator | 2.875 | $\Omega$ |
| Phase Inductance of Stator | $8.5 \times 10^{-3}$ | H |
| Voltage Constant ($V_{L\text{-}L}$) | 146.6077 | V/krpm |

controller ($K_p$ = 0.999667073 and $K_i$ = 82.62385185), which makes the system unstable. However, the settling time is low; therefore, this row is highlighted in bold is selected.

## 5.4 Analysis of the results of PSO

The performance of PSO is evaluated for different numbers of search agents for a fixed number of iterations, as displayed in Fig 6.

In Table 5, the value of cost function is minimum for $K_p$ = 0.852200229 and $K_i$ = 63.22674407 at 50 search agents, but the value of overshoot is higher. Nevertheless, due to lower settling time (i.e., < 0.02 s), the system rapidly achieve the steady state. Therefore, the row of 50 search agents (with bold digits) with a trade-off for overshoot and settling time is the best choice having a low value of error.

**5.4.1 PSO with inertial weight factor.** The performance of PSO_w (incorporated inertial weight in PSO) is evaluated for different numbers of search agents for a fixed number of iterations, as displayed in Fig 7. The value of the cost function is minimal for Kp = 0.343524674

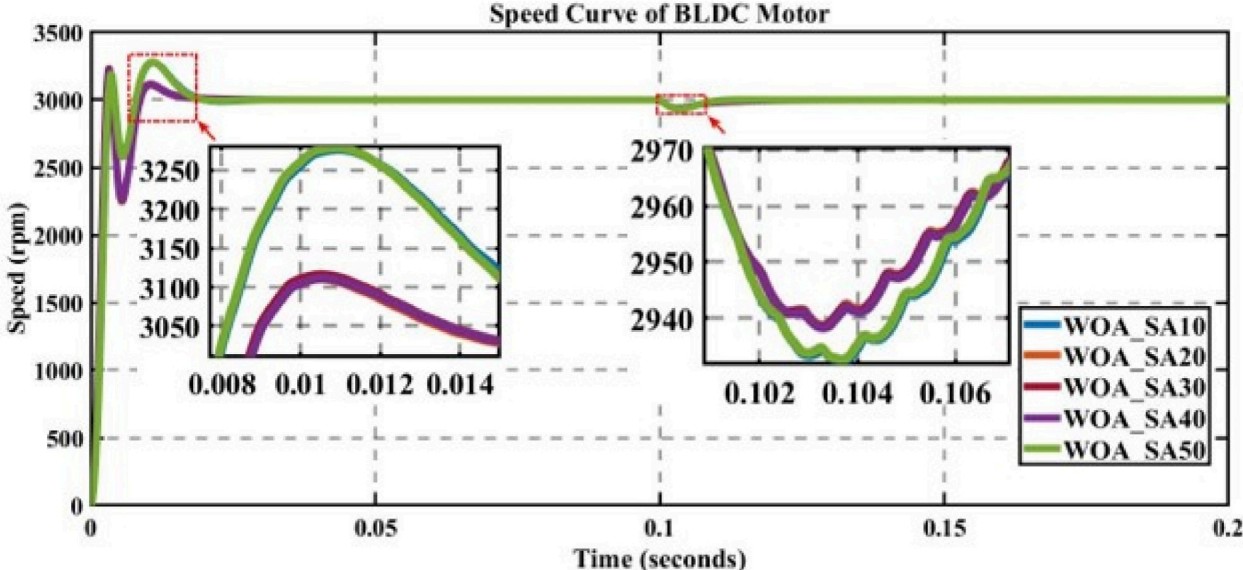

**Fig 3. Curve of sensorless BLDC motor's speed controlled by WOA-PI for different search agents.**

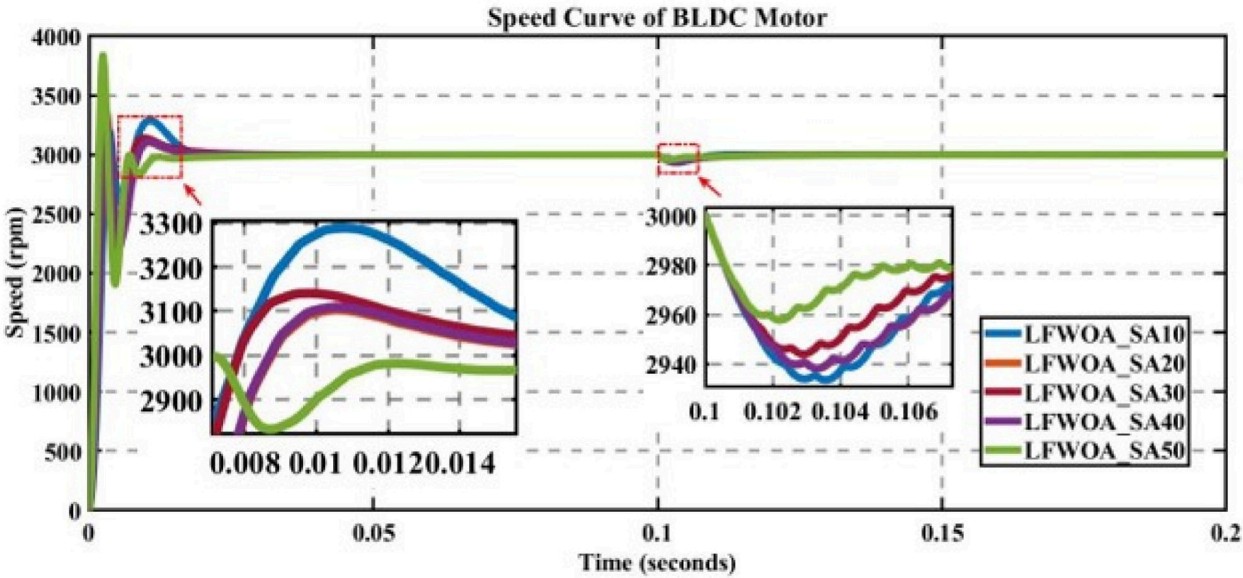

**Fig 4. Curve of sensorless BLDC motor's speed controlled by LFWOA-PI for different search agents.**

and Ki = 96.44265077 at 50 search agents in Table 6, but against this value, the value of overshoot is very high, causing instability. However, due to the shorter settling time, it is the best choice because it has a low value of error.

## 5.5 Analysis of the results of APSO

The performance of APSO is evaluated for different numbers of search agents for a fixed number of iterations, as displayed in Fig 8.

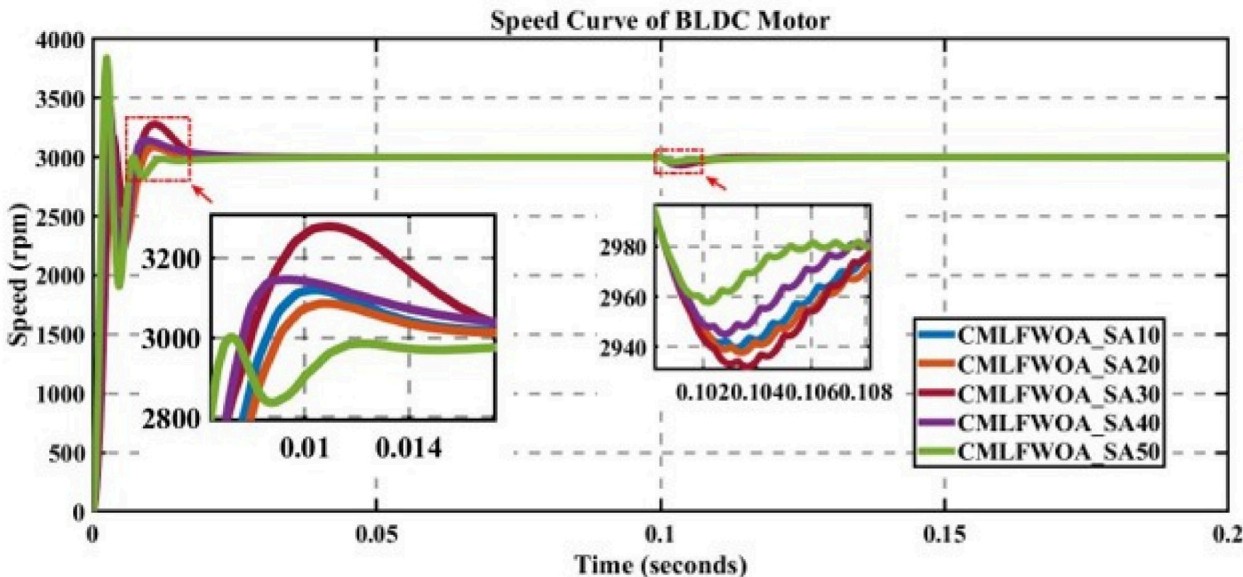

**Fig 5. Curve of sensorless BLDC motor's speed controlled by CMLFWOA-PI for different search agents.**

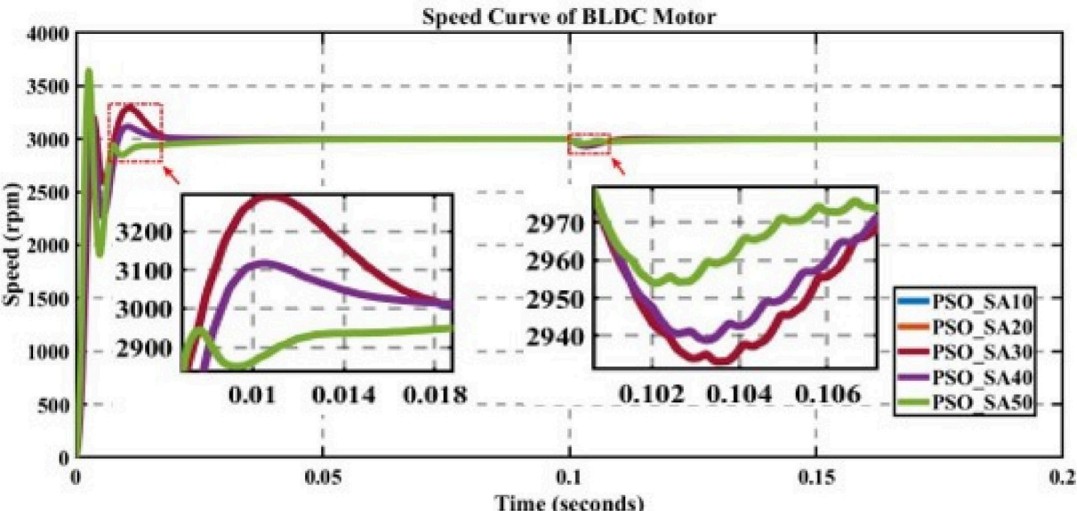

**Fig 6. Curve of sensorless BLDC motor's speed controlled by PSO-PI for different search agents.**

Table 7 shows that for 50 search agents against $K_p$ = 0.910522769 and $K_i$ = 70.24444048, the value of the cost function is minimal, though the value of overshoot makes the system unstable, but after reaching settling time, the steady-state is achieved; therefore, the highlighted 50 search agents' row in bold is the best choice, having a low value of error and an acceptable settling time.

### 5.6 Analysis of the results of ACO

The performance of ACO is evaluated for different numbers of search agents for a fixed number of iterations, as displayed in Fig 9, and for transient analysis, as observed in Table 8.

Table 8 shows that for 50 search agents for Kp = 0.979818182 and Ki = 79.39393939, the value of the cost function is minimal. Though, overshoot is really very high put due to small

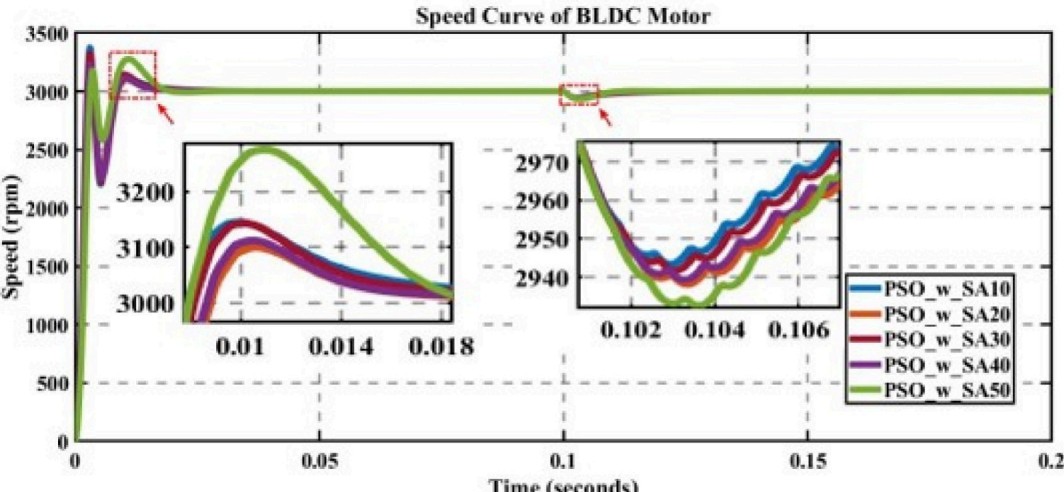

**Fig 7. Curve of sensorless BLDC motor's speed controlled by PSO_w-PI for different search agents.**

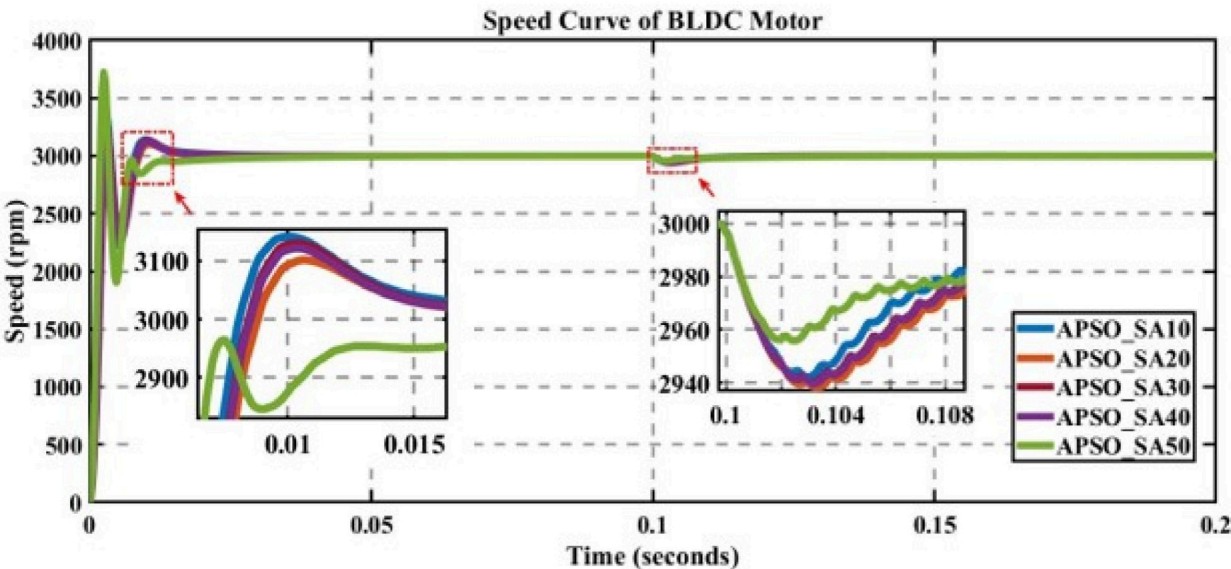

**Fig 8. Curve of sensorless BLDC motor's speed controlled by APSO for different search agents.**

settling time, this overshoot in transient time of the wave has eliminated; therefore, 50 search agents with a trade-off for overshoot and settling time is the best choice having minimal optimum value of ISE (highlighted in bold digits).

## 5.7 Comparison of algorithms

The performance of each algorithm is compared for the best-selected curve of each algorithm for some fixed iterations is presented in Fig 10.

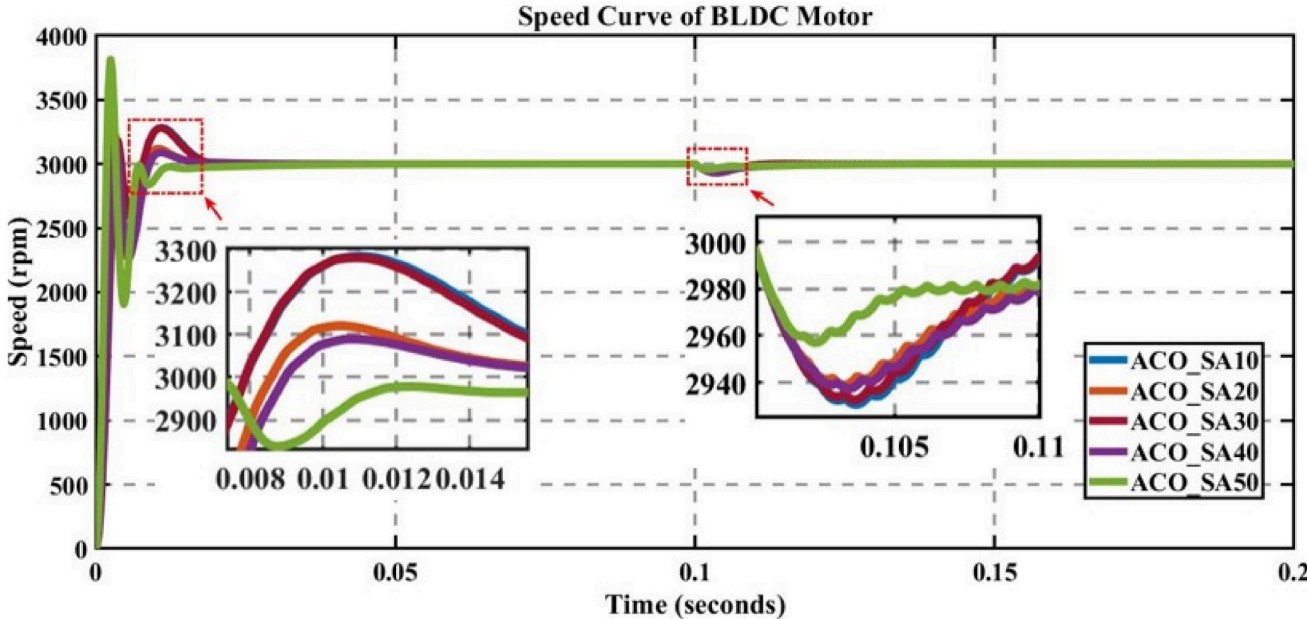

**Fig 9. Curve of sensorless BLDC motor's speed controlled by ACO for different search agents.**

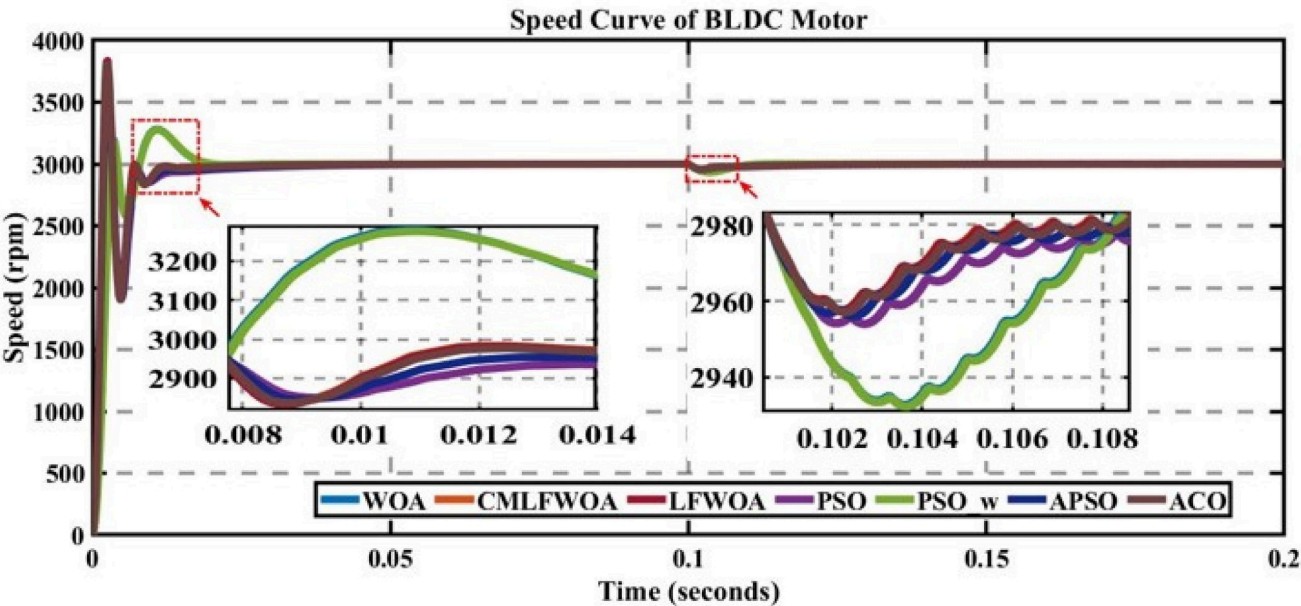

**Fig 10. Curve of sensorless BLDC motor's speed controlled by metaheuristic-PI controllers.**

**Table 2. Dynamic analysis of WOA for different population sizes.**

| No. of Iterations | No. of Search Agents | Rise Time $T_r$ | Settling Time $T_s$ | OS | Peak Time $T_p$ | Cost Function (ISE) |
|---|---|---|---|---|---|---|
| | | (s) | (s) | (%) | (s) | |
| 30 | 10 | 0.00193 | 0.014673 | 6.050 | 0.01096 | 0.312124 |
| 30 | 20 | 0.001714 | 0.011823 | 8.152 | 0.003188 | 0.309654 |
| 30 | 30 | 0.001744 | 0.01094 | 6.989 | 0.003204 | 0.309405 |
| 30 | 40 | 0.001752 | 0.011241 | 6.989 | 0.003256 | 0.309648 |
| **30** | **50** | **0.00192** | **0.01451** | **6.333** | **0.01088** | **0.283330** |

**Table 3. Dynamic analysis of LFWOA for different population sizes.**

| No. of Iterations | No. of Search Agents | Rise Time $T_r$ | Settling Time $T_s$ | OS | Peak Time $T_p$ | Cost Function (ISE) |
|---|---|---|---|---|---|---|
| | | (s) | (s) | (%) | (s) | |
| 30 | 10 | 0.001904 | 0.014176 | 6.610 | 0.0107 | 0.315262 |
| 30 | 20 | 0.001747 | 0.010926 | 6.989 | 0.003216 | 0.311562 |
| 30 | 30 | 0.001564 | 0.012682 | 13.068 | 0.003013 | 0.313145 |
| 30 | 40 | 0.001756 | 0.011443 | 6.989 | 0.003256 | 0.312376 |
| **30** | **50** | **0.00116** | **0.009641** | **27.564** | **0.002534** | **0.288097** |

Table 9 presents that the best optimum value of the cost function is approximately 0.283330, which is given in the result of the WOA for 30 search agents for $K_p$ = 0.348422343 and $K_i$ = 97.82639568 with a trade-off between settling time, overshoot, and rise time.

**Table 4. Dynamic analysis of CMLFWOA for different population sizes.**

| No. of Iterations | No. of Search Agents | Rise Time $T_r$ | Settling Time $T_s$ | OS | Peak Time $T_p$ | Cost Function (ISE) |
|---|---|---|---|---|---|---|
| | | (s) | (s) | (%) | (s) | |
| 30 | 10 | 0.001701 | 0.011073 | 8.152 | 0.003195 | 0.315995 |
| 30 | 20 | 0.001785 | 0.010845 | 5.851 | 0.003279 | 0.316919 |
| 30 | 30 | 0.001942 | 0.014591 | 5.897 | 0.01096 | 0.315234 |
| 30 | 40 | 0.001526 | 0.015032 | 14.368 | 0.002944 | 0.314915 |
| **30** | **50** | **0.001159** | **0.009641** | **27.564** | **0.002534** | **0.288583** |

**Table 5. Dynamic analysis of PSO for different population sizes.**

| No. of Iterations | No. of Search Agents | Rise Time $T_r$ | Settling Time $T_s$ | OS | Peak Time $T_p$ | Cost Function (ISE) |
|---|---|---|---|---|---|---|
| | | (s) | (s) | (%) | (s) | |
| 30 | 10 | 0.001909 | 0.014129 | 6.518 | 0.0107 | 0.314966 |
| 30 | 20 | 0.001906 | 0.014131 | 6.585 | 0.01081 | 0.314230 |
| 30 | 30 | 0.001914 | 0.014124 | 6.437 | 0.01084 | 0.313913 |
| 30 | 40 | 0.001749 | 0.011083 | 6.989 | 0.003236 | 0.311532 |
| **30** | **50** | **0.001279** | **0.017027** | **21.341** | **0.0026** | **0.287992** |

**Table 6. Dynamic analysis of PSO_w for different population sizes.**

| No. of Iterations | No. of Search Agents | Rise Time $T_r$ | Settling Time $T_s$ | OS | Peak Time $T_p$ | Cost Function (ISE) |
|---|---|---|---|---|---|---|
| | | (s) | (s) | (%) | (s) | |
| 30 | 10 | 0.001568 | 0.01351 | 13.068 | 0.002996 | 0.312681 |
| 30 | 20 | 0.001745 | 0.01116 | 6.989 | 0.003219 | 0.311544 |
| 30 | 30 | 0.001653 | 0.011777 | 10.556 | 0.003107 | 0.312803 |
| 30 | 40 | 0.001726 | 0.012271 | 8.152 | 0.003207 | 0.309078 |
| **30** | **50** | **0.001935** | **0.014674** | **6.029** | **0.01094** | **0.285883** |

**Table 7. Dynamic analysis of APSO for different population sizes.**

| No. of Iterations | No. of Search Agents | Rise Time $T_r$ | Settling Time $T_s$ | OS | Peak Time $T_p$ | Cost Function (ISE) |
|---|---|---|---|---|---|---|
| | | (s) | (s) | (%) | (s) | |
| 30 | 10 | 0.0016 | 0.01255 | 11.798 | 0.003053 | 0.313726 |
| 30 | 20 | 0.001753 | 0.011684 | 6.989 | 0.003241 | 0.312947 |
| 30 | 30 | 0.001707 | 0.011222 | 8.152 | 0.003176 | 0.309626 |
| 30 | 40 | 0.001712 | 0.011361 | 8.152 | 0.003187 | 0.309664 |
| **30** | **50** | **0.001224** | **0.010422** | **24.375** | **0.002534** | **0.287086** |

## 5.8 Results of statistical analysis

Statistical tests are performed to show any statistical difference between the performance of all these algorithms for solving this problem. Table 10 presents the statistics of data for the best-selected curve of each algorithm.

**Table 8. Dynamic analysis of ACO for different population sizes.**

| No. of Iterations | No. of Search Agents | Rise Time $T_r$ | Settling Time $T_s$ | OS | Peak Time $T_p$ | Cost Function (ISE) |
|---|---|---|---|---|---|---|
| | | (s) | (s) | (%) | (s) | |
| 30 | 10 | 0.001983 | 0.014568 | 5.954 | 0.01106 | 0.318023 |
| 30 | 20 | 0.001712 | 0.011738 | 8.152 | 0.003188 | 0.313860 |
| 30 | 30 | 0.001931 | 0.014484 | 6.069 | 0.01094 | 0.315153 |
| 30 | 40 | 0.001803 | 0.012055 | 5.851 | 0.003301 | 0.314984 |
| **30** | **50** | **0.001168** | **0.009518** | **27.564** | **0.002534** | **0.289149** |

**Table 9. Dynamic analysis of algorithms to select the best one.**

| Algorithms | No. of Search Agents | No. of Iterations | Rise Time $T_r$ | Settling Time $T_s$ | OS | Peak Time $T_p$ | Cost Function (ISE) |
|---|---|---|---|---|---|---|---|
| | | | (s) | (s) | % | (s) | |
| **WOA** | **30** | **50** | **0.00192** | **0.01451** | **6.333** | **0.01088** | **0.283330** |
| CMLFWOA | 30 | 50 | 0.001159 | 0.009641 | 27.564 | 0.002534 | 0.288583 |
| LFWOA | 30 | 50 | 0.00116 | 0.009641 | 27.564 | 0.002534 | 0.288097 |
| PSO | 30 | 50 | 0.001279 | 0.017027 | 21.341 | 0.0026 | 0.287992 |
| PSO_w | 30 | 50 | 0.001935 | 0.014674 | 6.029 | 0.01094 | 0.285883 |
| APSO | 30 | 50 | 0.001224 | 0.010422 | 24.375 | 0.002534 | 0.287086 |
| ACO | 30 | 50 | 0.001168 | 0.009518 | 27.564 | 0.002534 | 0.289149 |

**Table 10. Descriptive statistics of algorithms.**

| Algorithms | N | Mean | Std. Deviation | Std. Error | 95% Confidence Interval for Mean | | Min | Max |
|---|---|---|---|---|---|---|---|---|
| | | | | | Lower Bound | Upper Bound | | |
| WOA | 50 | .28824645 | .001507125 | .000213140 | .28781813 | .28867477 | .283330 | .290516 |
| AC0 | 50 | .29424223 | .003504530 | .000495615 | .29324625 | .29523821 | .289149 | .301229 |
| APSO | 50 | .29002017 | .002536266 | .000358682 | .28929938 | .29074097 | .287086 | .298179 |
| CMLFWOA | 50 | .29370596 | .003275075 | .000463166 | .29277520 | .29463673 | .288583 | .299318 |
| PSO | 50 | .28965501 | .001512479 | .000213897 | .28922516 | .29008485 | .287992 | .295715 |
| PSO_w | 50 | .28873002 | .001185853 | .000167705 | .28839300 | .28906703 | .285883 | .292458 |
| LFWOA | 50 | .28976726 | .001901983 | .000268981 | .28922672 | .29030780 | .288097 | .296244 |

**Table 11. Group statistics of APSO vs. WOA.**

| Algorithms | N | Mean | Std. Deviation | Std. Error Mean |
|---|---|---|---|---|
| APSO | 50 | .29002017 | .002536266 | .000358682 |
| WOA | 50 | .28824645 | .001507125 | .000213140 |

In Tables 11 to 56, results for different statistical tests (i.e., T-test, ANOVA (One-Way), Mann Whitney U test, Wilcoxon signed-rank test, Friedman test, and Friedman aligned ranks test) are shown in which the test variable is an error, ISE values for APSO (Search Agents = 50), ACO (Search Agents = 50), PSO (Search Agents = 50), PSO_w (Search Agents = 50), CMLFWOA (Search Agents = 50), and LWOA (Search Agents = 50) are compared with the ISE values of WOA (Search Agents = 50) as shown in Table 9. These statistical tests are

**Table 12. T-test for APSO vs. WOA.**

| | Levene's Test for Equality of Variances | | T-Test for Equality of Means | | | | | | |
| | F | Sig. | t | df | Sig. (2-tailed) | Mean Difference | Std. Error Difference | 95% Confidence Interval of Difference | |
| | | | | | | | | Lower | Upper |
|---|---|---|---|---|---|---|---|---|---|
| Equal Variances assumed | 11.78 | .001 | 4.251 | 98 | .000 | .001773723 | .000417231 | .000945742 | .002601704 |
| Equal Variances not assumed | | | 4.251 | 79.768 | .000 | .001773723 | .000417231 | .000943371 | .002604076 |

**Table 13. Group statistics of ACO vs. WOA.**

| Algorithms | N | Mean | Std. Deviation | Std. Error Mean |
|---|---|---|---|---|
| ACO | 50 | .29424223 | .003504530 | .000495615 |
| WOA | 50 | .28824645 | .001507125 | .000213140 |

**Table 14. T-test for ACO vs. WOA.**

| | Levene's Test for Equality of Variances | | T-Test for Equality of Means | | | | | | |
| | F | Sig. | t | df | Sig. (2-tailed) | Mean Difference | Std. Error Difference | 95% Confidence Interval of Difference | |
| | | | | | | | | Lower | Upper |
|---|---|---|---|---|---|---|---|---|---|
| Equal Variances assumed | 41.86 | .000 | 11.11 | 98 | .000 | .005995779 | .000539503 | .004925153 | .007066404 |
| Equal Variances not assumed | | | 11.11 | 66.525 | .000 | .005995779 | .000539503 | .004918786 | .007072772 |

**Table 15. Group statistics of PSO vs. WOA.**

| Algorithms | N | Mean | Std. Deviation | Std. Error Mean |
|---|---|---|---|---|
| PSO | 50 | .28965501 | .001512479 | .000213897 |
| WOA | 50 | .28824645 | .001507125 | .000213140 |

**Table 16. T-test for PSO vs. WOA.**

| | Levene's Test for Equality of Variances | | T-Test for Equality of Means | | | | | | |
| | F | Sig. | t | df | Sig. (2-tailed) | Mean Difference | Std. Error Difference | 95% Confidence Interval of Difference | |
| | | | | | | | | Lower | Upper |
|---|---|---|---|---|---|---|---|---|---|
| Equal Variances assumed | .016 | .899 | 4.665 | 98 | .000 | .001408555 | .000301961 | .000809324 | .002007787 |
| Equal Variances not assumed | | | 4.665 | 97.999 | .000 | .001408555 | .000301961 | .000809324 | .002007787 |

**Table 17. Group statistics of PSO_w vs. WOA.**

| Algorithms | N | Mean | Std. Deviation | Std. Error Mean |
|---|---|---|---|---|
| PSO_w | 50 | .28873002 | .001185853 | .000167705 |
| WOA | 50 | .28824645 | .001507125 | .000213140 |

**Table 18. T-test for PSO_w vs. WOA.**

| | Levene's Test for Equality of Variances | | T-Test for Equality of Means | | | | | | 95% Confidence Interval of Difference | |
|---|---|---|---|---|---|---|---|---|---|---|
| | F | Sig. | t | df | Sig. (2-tailed) | Mean Difference | Std. Error Difference | | Lower | Upper |
| Equal Variances assumed | .657 | .420 | 1.783 | 98 | .078 | .000483564 | .000271207 | | -.00005464 | .001021766 |
| Equal Variances not assumed | | | 1.783 | 92.861 | .078 | .000483564 | .000271207 | | -.00005501 | .001022139 |

**Table 19. Group statistics of CMLFWOA vs. WOA.**

| Algorithms | N | Mean | Std. Deviation | Std. Error Mean |
|---|---|---|---|---|
| CMLFWOA | 50 | .29370596 | .003275075 | .000463166 |
| WOA | 50 | .28824645 | .001507125 | .000213140 |

**Table 20. T-test for CMLFWOA vs. WOA.**

| | Levene's Test for Equality of Variances | | T-Test for Equality of Means | | | | | | 95% Confidence Interval of Difference | |
|---|---|---|---|---|---|---|---|---|---|---|
| | F | Sig. | t | df | Sig. (2-tailed) | Mean Difference | Std. Error Difference | | Lower | Upper |
| Equal Variances assumed | 55.34 | .000 | 10.71 | 98 | .000 | .005459514 | .000509854 | | .004447725 | .006471302 |
| Equal Variances not assumed | | | 10.71 | 68.862 | .000 | .005459514 | .000509854 | | .004442347 | .006476680 |

**Table 21. Group statistics of LFWOA vs. WOA.**

| Algorithms | N | Mean | Std. Deviation | Std. Error Mean |
|---|---|---|---|---|
| LFWOA | 50 | .28976726 | .001901983 | .000268981 |
| WOA | 50 | .28824645 | .001507125 | .000213140 |

**Table 22. T-Test for LFWOA vs. WOA.**

| | Levene's Test for Equality of Variances | | T-Test for Equality of Means | | | | | | 95% Confidence Interval of Difference | |
|---|---|---|---|---|---|---|---|---|---|---|
| | F | Sig. | t | df | Sig. (2-tailed) | Mean Difference | Std. Error Difference | | Lower | Upper |
| Equal Variances assumed | 1.521 | .220 | 4.43 | 98 | .000 | .001520807 | .000343190 | | .000839758 | .002201856 |
| Equal Variances not assumed | | | 4.43 | 93.134 | .000 | .001520807 | .000343190 | | .000839313 | .002202301 |

**Table 23. Mean rank comparison between APSO vs. WOA.**

| Algorithms | N | Mean Rank | Sum of Ranks |
|---|---|---|---|
| APSO | 50 | 58.96 | 2948 |
| WOA | 50 | 42.04 | 2102 |
| Total | 100 | | |

**Table 24. Mann-Whitney U test between APSO vs. WOA.**

| | Error |
|---|---|
| Mann-Whitney U | 827 |
| Wilcoxon W | 2102 |
| Z | -2.916 |
| Asymp. Sig. (2-tailed) | .004 |

**Table 25. Mean rank comparison between ACO vs. WOA.**

| Algorithms | N | Mean Rank | Sum of Ranks |
|---|---|---|---|
| ACO | 50 | 73.72 | 3686 |
| WOA | 50 | 27.28 | 1364 |
| Total | 100 | | |

**Table 26. Mann-Whitney U test between ACO vs. WOA.**

| | Error |
|---|---|
| Mann-Whitney U | 89 |
| Wilcoxon W | 1364 |
| Z | -8.005 |
| Asymp. Sig. (2-tailed) | .000 |

**Table 27. Mean rank comparison between PSO vs. WOA.**

| Algorithms | N | Mean Rank | Sum of Ranks |
|---|---|---|---|
| PSO | 50 | 63.02 | 3151 |
| WOA | 50 | 37.98 | 1899 |
| Total | 100 | | |

**Table 28. Mann-Whitney U test between PSO vs. WOA.**

| | Error |
|---|---|
| Mann-Whitney U | 624 |
| Wilcoxon W | 1899 |
| Z | -4.316 |
| Asymp. Sig. (2-tailed) | .000 |

**Table 29. Mean rank comparison between PSO_w vs. WOA.**

| Algorithms | N | Mean Rank | Sum of Ranks |
|---|---|---|---|
| PSO_w | 50 | 52.64 | 2632 |
| WOA | 50 | 48.36 | 2418 |
| Total | 100 | | |

**Table 30. Mann-Whitney U test between PSO_w vs. WOA.**

| | Error |
|---|---|
| Mann-Whitney U | 1143 |
| Wilcoxon W | 2418 |
| Z | -.738 |
| Asymp. Sig. (2-tailed) | .461 |

**Table 31. Mean rank comparison between CMLFWOA vs. WOA.**

| Algorithms | N | Mean Rank | Sum of Ranks |
|---|---|---|---|
| CMLFWOA | 50 | 74.18 | 3709 |
| WOA | 50 | 26.82 | 1341 |
| Total | 100 | | |

**Table 32. Mann-Whitney U test between CMLFWOA vs. WOA.**

| | Error |
|---|---|
| Mann-Whitney U | 66 |
| Wilcoxon W | 1341 |
| Z | -8.162 |
| Asymp. Sig. (2-tailed) | .000 |

**Table 33. Mean rank comparison between LFWOA vs. WOA.**

| Algorithms | N | Mean Rank | Sum of Ranks |
|---|---|---|---|
| LFWOA | 50 | 61.94 | 3097 |
| WOA | 50 | 39.06 | 1953 |
| Total | 100 | | |

**Table 34. Mann-Whitney U test between LFWOA vs. WOA.**

| | Error |
|---|---|
| Mann-Whitney U | 678 |
| Wilcoxon W | 1953 |
| Z | -3.943 |
| Asymp. Sig. (2-tailed) | .000 |

**Table 35. Wilcoxon signed-rank test for APSO vs. WOA.**

|  | N | Mean Rank | Sum of Ranks |
|---|---|---|---|
| Negative Ranks | 13 | 19.54 | 254.00 |
| Positive Ranks | 37 | 27.59 | 1021.00 |
| Ties | 0 |  |  |
| Total | 50 |  |  |

**Table 36. Wilcoxon test statistics for APSO vs. WOA.**

|  | Test Statistics |
|---|---|
| Z | -3.702 |
| Asymp. Sig. (2-tailed) | 0.000214 |

**Table 37. Wilcoxon signed-rank test for ACO vs. WOA.**

|  | N | Mean Rank | Sum of Ranks |
|---|---|---|---|
| Negative Ranks | 3 | 3.67 | 11.00 |
| Positive Ranks | 47 | 26.89 | 1264.00 |
| Ties | 0 |  |  |
| Total | 50 |  |  |

**Table 38. Wilcoxon test statistics for ACO vs. WOA.**

|  | Test Statistics |
|---|---|
| Z | -6.048 |
| Asymp. Sig. (2-tailed) | $1.4686 \times 10^{-9}$ |

**Table 39. Wilcoxon signed-rank test for PSO vs. WOA.**

|  | N | Mean Rank | Sum of Ranks |
|---|---|---|---|
| Negative Ranks | 14 | 17.21 | 241.00 |
| Positive Ranks | 36 | 28.72 | 1034.00 |
| Ties | 0 |  |  |
| Total | 50 |  |  |

performed by using SPSS software. Each algorithm is run 50 times to collect the data. N in Table 10 represents the number of times the data for each algorithm is collected.

**5.8.1 Results of t-test.** Group statistics of APSO and WOA are mentioned in Table 11. The significance value (2-tailed) for the independent sample t-test in Table 12 is 0.000 (i.e., <

**Table 40. Wilcoxon test statistics for PSO vs. WOA.**

|  | Test Statistics |
|---|---|
| Z | -3.828 |
| Asymp. Sig. (2-tailed) | 0.000129 |

**Table 41. Wilcoxon signed-rank test for PSO_w vs. WOA.**

|  | N | Mean Rank | Sum of Ranks |
|---|---|---|---|
| Negative Ranks | 23 | 20.96 | 482.00 |
| Positive Ranks | 27 | 29.37 | 793.00 |
| Ties | 0 |  |  |
| Total | 50 |  |  |

**Table 42. Wilcoxon test statistics for PSO_w vs. WOA.**

|  | Test Statistics |
|---|---|
| Z | -1.501 |
| Asymp. Sig. (2-tailed) | 0.133 |

**Table 43. Wilcoxon signed-rank test for CMLFWOA vs. WOA.**

|  | N | Mean Rank | Sum of Ranks |
|---|---|---|---|
| Negative Ranks | 4 | 3.50 | 14.00 |
| Positive Ranks | 46 | 27.41 | 1261.00 |
| Ties | 0 |  |  |
| Total | 50 |  |  |

**Table 44. Wilcoxon test statistics CMLFWOA for vs. WOA.**

|  | Test Statistics |
|---|---|
| Z | -6.019 |
| Asymp. Sig. (2-tailed) | $1.7569 \times 10^{-9}$ |

**Table 45. Wilcoxon signed-rank test for LFWOA vs. WOA.**

|  | N | Mean Rank | Sum of Ranks |
|---|---|---|---|
| Negative Ranks | 13 | 16.62 | 216.00 |
| Positive Ranks | 37 | 28.62 | 1059.00 |
| Ties | 0 |  |  |
| Total | 50 |  |  |

**Table 46. Wilcoxon test statistics for LFWOA vs. WOA.**

|  | Test Statistics |
|---|---|
| Z | -4.069 |
| Asymp. Sig. (2-tailed) | 0.000047 |

**Table 47. One way ANOVA test.**

|  | Sum of Squares | df | Mean Square | F | Sig. |
|---|---|---|---|---|---|
| Between Groups | .002 | 6 | .000 | 50.626 | .000 |
| Within Groups | .002 | 343 | .000 |  |  |

**Table 48. Post hoc tests (Tukey HSD).**

| | (I) Algorithms | (J) Algorithms | Mean Difference (I-J) | Std. Error | Sig. | 95% Confidence Interval | |
|---|---|---|---|---|---|---|---|
| | | | | | | Lower | Upper |
| Tukey HSD | WOA | ACO | -.005995779* | .000472218 | .000 | -.00739635 | -.00459521 |
| | | APSO | -.001773723* | .000472218 | .004 | -.00317429 | -.00037315 |
| | | CMLFWOA | -.005459514* | .000472218 | .000 | -.00686008 | -.00405894 |
| | | PSO | -.001408555 | .000472218 | .048 | -.00280912 | -.00000799 |
| | | PSO_w | -.000483564* | .000472218 | .948 | -.00188413 | .00091700 |
| | | LFWOA | -.001520807* | .000472218 | .023 | -.00292138 | -.00012024 |

**Table 49. Post hoc tests (Scheffe).**

| | (I) Algorithms | (J) Algorithms | Mean Difference (I-J) | Std. Error | Sig. | 95% Confidence Interval | |
|---|---|---|---|---|---|---|---|
| | | | | | | Lower | Upper |
| Scheffe | WOA | ACO | -.005995779* | .000472218 | .000 | -.00768195 | -.00430961 |
| | | APSO | -.001773723* | .000472218 | .031 | -.00345989 | -.00008755 |
| | | CMLFWOA | -.005459514* | .000472218 | .000 | -.00714569 | -.00377334 |
| | | PSO | -.001408555 | .000472218 | .183 | -.00309473 | .00027762 |
| | | PSO_w | -.000483564 | .000472218 | .948 | -.00216974 | .00120261 |
| | | LFWOA | -.001520807 | .000472218 | .113 | -.00320698 | .00016536 |

**Table 50. Post hoc tests (Bonferroni).**

| | (I) Algorithms | (J) Algorithms | Mean Difference (I-J) | Std. Error | Sig. | 95% Confidence Interval | |
|---|---|---|---|---|---|---|---|
| | | | | | | Lower | Upper |
| Bonferroni | WOA | ACO | -.005995779* | .000472218 | .000 | -.00744118 | -.00455037 |
| | | APSO | -.001773723* | .000472218 | .004 | -.00321913 | -.00032832 |
| | | CMLFWOA | -.005459514* | .000472218 | .000 | -.00690492 | -.00401411 |
| | | PSO | -.001408555 | .000472218 | .064 | -.00285396 | .00003685 |
| | | PSO_w | -.000483564 | .000472218 | 1.000 | -.00192897 | .00096184 |
| | | LFWOA | -.001520807* | .000472218 | .029 | -.00296621 | -.00007540 |

0.05), so there is a significant difference in the performance of these algorithms. In Table 12, the value of F is very important as this value is calculated as the ratio variance in between defined groups of data to the variance within each group. The large value of F (as mentioned in Table 12) indicates large difference between groups. Moreover, WOA has smaller value of mean in Table 12 which indicates that WOA is superior to APSO.

**Table 51. Friedman test.**

| Algorithms | Mean Rank |
|---|---|
| WOA | 2.40 |
| ACO | 5.94 |
| APSO | 3.64 |
| CMLFWOA | 5.96 |
| PSO | 3.64 |
| PSO_w | 2.60 |
| LFWOA | 3.82 |

**Table 52. Friedman test (Test statistics).**

| Algorithms | Mean Rank |
|---|---|
| N | 50 |
| Chi-Square | 133.037 |
| df | 6 |
| Asymp. Sig. | < 0.001 |

**Table 53. p-values of comparison of algorithms with WOA.** (arranged in ascending order).

| Comparisons | ACO-WOA | CMLFWOA-WOA | LFWOA-WOA | PSO-WOA | APSO-WOA | PSO_w-WOA |
|---|---|---|---|---|---|---|
| p-Value | $1.4686 \times 10^{-9}$ | $1.7569 \times 10^{-9}$ | 0.000047 | 0.000129 | 0.000214 | 0.133334 |

**Table 54. Friedman aligned ranks test.**

| Aligned Algorithms | Mean Rank |
|---|---|
| Aligned WOA | 3.73 |
| Aligned ACO | 4.05 |
| Aligned APSO | 4.05 |
| Aligned CMLFWOA | 4.10 |
| Aligned PSO | 3.87 |
| Aligned PSO_w | 4.00 |
| Aligned LFWOA | 4.20 |

**Table 55. Friedman aligned ranks test (Test statistics).**

| Algorithms | Mean Rank |
|---|---|
| N | 50 |
| Chi-Square | 1.560 |
| df | 6 |
| Asymp. Sig. | 0.955 |

**Table 56. Q-value of comparison of algorithms with WOA.**

| Comparisons | ACO-WOA | CMLFWOA-WOA | LFWOA-WOA | PSO-WOA | APSO-WOA | PSO_w-WOA |
|---|---|---|---|---|---|---|
| Q-Value | 0.8 | 0.8 | 0.925 | 0.35 | 0.675 | 1.175 |

Table 13 gives information about the statistics of ACO and WOA. The significance value (2-tailed) of the t-test for the comparison of ACO and WOA in Table 14 is .000 (i.e., < 0.05), as the performance of these algorithms is significantly unique. From Table 18 the mean value for WOA is smaller; therefore, WOA has performed better than ACO.

In Table 15, the statistics of PSO and WOA are presented. The value of mean for WOA (i.e., .28824645) is less than the mean of PSO (i.e., .28965501) which shows that WOA is better than PSO. There is significantly no difference between the performance of PSO and WOA as the significance value (2-tailed) of the t-test for the comparison of PSO and WOA is .000, which is less than 0.05, as shown in Table 16.

Table 17 gives information about group statistics of PSO with inertial weight and WOA is presented. In Table 18 Sig. (2-tailed) i.e., 0.078 of the t-test is greater than 0.05 (for 95% confidence interval difference), which indicates no significant difference in the performance of PSO_w and WOA. The mean value of PSO_w in Table 18 is greater than the mean value of WOA; therefore, WOA is better than PSO_w.

The group statistics of CMLFWOA and WOA are presented in Table 19. The performance of WOA and CMLFWOA is different according to the significant value, i.e., 0.000 and the large value of F of the t-test's statistics in Table 20. The value of mean of WOA is less than CMLFWOA; therefore, the canonical version of WOA is the better option.

Table 21 displays the group statistics of LFWOA and WOA. WOA is more efficient than LFWOA because mean of WOA is less than the mean of LFWOA. The significant value (2-tailed) is .000 in Table 22, which shows that the performance of WOA is significantly different from the performance of LFWOA (for a 95% confidence interval difference). WOA is efficient because of the smaller value of mean.

Hence, the value of significance level (2-tailed) is less than 0.05 except for PSO_w. But overall on the basis of performance, WOA is the best choice for solving this certain problem of optimization.

**5.8.2 Results of Mann-Whitney U test.** Tables 23 and 24 the mean rank comparison of ACO vs. WOA and test statistics of Mann-Whitney U test are given. In Table 23, the mean rank of WOA is less than the mean rank value of APSO, which shows that the performance of WOA is better than APSO algorithm in finding the global best for regulating parameters of PI controller. In Table 24 the negative value of Z indicates that WOA performs well because these test statistics are calculated for APSO-WOA. The significance level for WOA vs. APSO is also less than 0.05 which shows a statistical significant difference between the performances of both algorithms.

Table 25 displays that the mean rank of WOA is 27.28 which is less than the mean rank value of ACO. The lower value of mean rank shows that the performance of WOA is better than ACO in finding the global best values of the gains of the speed controller. In Table 26 the value of (2-tailed) significance level is .000, i.e., < 0.05 this statistical value presents a significant performance difference between ACO and WOA.

In Table 27, the mean rank of WOA is less (i.e., 37.98) than the value of mean rank of PSO (i.e., 63.02). The lower value of mean rank shows that WOA is superior to PSO in finding the global best optimum solution (as the value of ISE is minimum). In Table 27 the lower mean rank of WOA and negative value of Z in Table 28 indicates WOA is better than PSO. Another test statistics of Mann-Whitney U test in Table 28, i.e., value of significance level indicates a significant difference between the performances of both algorithms.

The mean rank comparison and test statistics of Mann-Whitney U tests are presented in Tables 29 and 30, respectively. In Table 29, the mean rank of WOA is less than the mean rank of PSO_w. The lower value of mean rank shows that the dominance of the performance of WOA is more efficient in finding the global best. The value of significance level is .461 which is greater than 0.05 due to which the null hypothesis of no difference between the performance of both algorithms is retained in the case of PSO_w vs. WOA.

Table 31 presents the mean rank comparison of WOA and CMLFWOA. The mean rank value of WOA is less than the mean rank value of CMLFWOA; therefore, the performance of WOA in finding the optimum solution (globally), is better than CMLFWOA. In Table 32, negative Z-value also claims the superiority over CMLFWOA. The value of 2-tailed significance level is .000 which statistically shows significant difference between the performances of WOA and CMLFWOA.

The mean rank of WOA (i.e., 39.06) is presented in Table 33 which is less than the mean rank value of LFWOA (i.e., 61.94). Lower mean rank in Table 33 and negative value of Z (-3.943) in Table 34 shows that WOA performs better than LFWOA. The value of Asymp. Sig. (2-tailed) (i.e., significance level for LFWOA vs. WOA) is 0.000047. This significance level is less than 0.05 which presents a significant difference between the performances of WOA and LFWOA.

The results of Mann-Whitney U test show that WOA is most efficient algorithm in finding the global optimal solution of this certain problem. Statistically there is a difference among WOA and other algorithms, except PSO_w.

**5.8.3 Results for Wilcoxon signed-rank test.** Tables 35 and 36 provide the results of Wilcoxon signed-rank test of other algorithms vs. WOA. In Table 36 the negative Z-value suggests the performance supremacy of WOA over APSO. The Wilcoxon signed-rank test is performed for APSO-WOA; therefore, the value of Z has negative sign. Moreover, the associated p-value (Asymp. Sig. (2-tailed)) is also less than 0.05 which represents a significant statistical difference between APSO and WOA in error minimization. In Table 35 the large number of positive ranks also depicts that WOA is efficient than APSO.

The test statistics for ACO-WOA in Wilcoxon signed-rank test are shown in Table 38 in which the negative Z-value suggests the performance supremacy of WOA. In Table 38 the p-value (Asymp. Sig. (2-tailed)) is less than 0.05 and indicates a significant statistical difference between the performances of ACO and WOA. In Table 37 the number of positive ranks is larger than the negative ranks; therefore, the efficiency of WOA is higher than ACO.

Table 40 presents the test statistics of Wilcoxon signed-rank test for PSO-WOA, the negative Z-value suggests the performance supremacy of WOA and the p-value (Asymp. Sig. (2-tailed)), i.e., 0.000129 < 0.05 which indicates a statistical difference between the performances of PSO and WOA. The larger value of positive ranks number in Table 39 shows that WOA performs better than PSO.

Table 41 presents mean rank and sum of rank of the negative ranks and the positive ranks for Wilcoxon signed-rank Test of PSO_w-WOA. Number of positive ranks is larger than the negative ranks; therefore, WOA is efficient than PSO_w. In Table 42 the test statistics (Z-value and Asymp. Sig. (2-tailed), i.e., p-value (0.133) which is > 0.05) of Wilcoxon signed-rank test suggests the performance supremacy of WOA over PSO_w but there is statistically no significant difference between their performance.

In Table 43 the larger number of positive ranks presents the supremacy of WOA in finding the optimal solution. In Table 44 the test statistics of Wilcoxon signed-rank test for CMLFWOA-WOA, the negative Z-value presents the performance supremacy of WOA and the p-value (Asymp. Sig. (2-tailed)), i.e., $1.7569 \times 10^{-9} < 0.05$ which statistically shows a significant difference between the performances of both algorithms.

Table 45 presents the mean rank and sum of ranks of positive ranks and negative ranks. The large number of positive ranks shows that WOA performs better than LFWOA. Table 46 presents the test statistics of Wilcoxon signed-rank test for LFWOA-WOA. The negative Z-value presents the performance supremacy of WOA and the p-value (Asymp. Sig. (2-tailed)), i.e., 0.000047 < 0.05 which statistically shows a significant difference between the performances of both algorithms.

**5.8.4 Results of one-way ANOVA.** The results in Table 47 specify that the significant value is .000, so the performance of these metaheuristic techniques is significantly distinct. The larger value of F depicts a difference in variation among groups relative to the within-group variation. In the ANOVA test, each subject is compared multiple times under different conditions; therefore, to identify the difference between groups from each other, post hoc tests Tables 47–50 are performed. Due to multiple comparisons, it is hectic to find which algorithm

is different from the other in Table 48 Tukey HSD (Honestly Significant Difference), which helps to find a comparison of individual algorithm. In Table 49, the Scheffe (Post Hoc) test shows the individual comparison of each algorithm with others if the condition of variance homogeneity is violated. In Table 50, the Bonferroni Correction is performed to prevent the possibility of the occurrence of type 1 error (rejecting the true null hypothesis), which happens due to multiple comparisons. The flagged values with "*" in Tables 48 to 50 represent which algorithm significantly differs from WOA according to specific statistical tests.

**5.8.5 Results of Friedman test.** The result of the Friedman test is show in Table 51. In this table the mean rank of each algorithm is provided. The value of mean rank for WOA is lowest which indicates that WOA is the most efficient algorithm than other algorithms. Table 52 presents the test statistics of the Friedman test, N is the size of data set which is ultimately the number of times each algorithm is run. In Table 52 the high value of chi-square, i.e., 133.037 shows significant difference between the performances of algorithms. The value of df (degree of freedom) presents the number of comparisons those are made independently. The Asymp. Sig. < 0.001 rejects the null hypothesis and shows a significant difference statistically between the performance of algorithms. In case of the Friedman test similar to the ANOVA test, due to multiple comparison there are chances of the occurrence of the type 1 error. Therefore, post hoc tests are applied to avoid the possibility of these errors. Holm's Step-Down and Bonferroni-Dunn corrections are applied which promote the accuracy in results, and tell which algorithm (group) is statistically different from other. After Table 52 these corrections are applied.

The selected significance level $\alpha$ is 0.05 and total number of comparisons (k) is 6. Following are the steps for Holm's adjustment and Bonferroni correction to show the individual difference of each algorithm from WOA and avoid the possibility of type 1 error.

**Holm's Step-Down (Post Hoc test):** The criteria which is followed in Holm's adjustment procedure is $\frac{\alpha}{k+1-i}$ (here i is the rank number of arranged p-value (presented in Wilcoxon signed-rank test statistics) of each comparison in ascending order as shown in Table 53. Using this relation of the Holm's adjustment the following level of significance value is set for each comparison.

1. Holm's adjustment for ACO-WOA (smallest p-value) is $\frac{0.05}{6} = 0.008333$. The p-value for ACO-WOA in Table 53 i.e., $1.4686 \times 10^{-9}$ which is less than 0.008333 that statistically indicates a significant difference between the performance of ACO and WOA.

2. Holm's adjustment for CMLFWOA-WOA is $\frac{0.05}{5} = 0.01$. In Table 53 p-value for CMLFWOA-WOA is $1.7569 \times 10^{-9}$ which is less than 0.01. Therefore, statistically there is a significant difference between the performance of CMLFWOA and WOA.

3. Holm's adjustment for the comparison, LFWOA-WOA is $\frac{0.05}{4} = 0.0125$. From Table 53 the p-value corresponds to the LFWOA-WOA comparison is 0.000047, i.e., < 0.0125 and presents a significant difference between the performance of LFWOA and WOA.

4. Holm's adjustment for PSO-WOA is $\frac{0.05}{3} = 0.016667$. Comparison of the p-value for PSO-WOA (from Table 53) i.e., 0.000129 which is less than 0.016667. It shows statistically a significant difference between the performance of PSO and WOA.

5. Holm's adjustment for APSO-WOA is $\frac{0.05}{2} = 0.025$. The p-value of APSO-WOA comparison in Table 53 is 0.000214 which is less than 0.025 that statistically indicates a significant difference between the performance of APSO and WOA.

6. Holm's adjustment for PSO_w-WOA (largest p-value) is $\frac{0.05}{1} = 0.05$. The p-value for the comparison of PSO_w-WOA in Table 53 i.e., 0.133334 which is greater than 0.05. This

p-value indicates that there is statistically no significant difference between the performance of PSO_w and WOA.

**Bonferroni-Dunn (Post Hoc test):** This correction is helpful in avoiding the possibility of type 1 error due to multiple comparison on same data set. The decision rule for selecting the significance level for each comparison is done by using a new significance threshold, i.e., $\frac{z}{k}$. In this case the new adjusted threshold is 0.008333, now the decision about each comparison depends on this value. By comparing each p-value in Table 53, it is found that there is a significant difference between ACO and WOA, CMLFWOA and WOA, LFWOA and WOA, PSO and WOA, and APSO and WOA except PSO_w and WOA. Because 0.133334 > 0.008333 which statistically shows no difference among the performances of PSO_w and WOA techniques.

**5.8.6 Results of Friedman aligned ranks test.** The results of Friedman aligned ranks test are shown in Tables 54 and 55. Table 54 shows that the value of mean rank for aligned WOA is lowest that suggests that WOA gives the best solution. The value of significance level (Asymp. Sig.) is 0.955, i.e., > 0.05. Therefore, it is found that according to Friedman aligned ranks test the null hypothesis is not rejected.

The following post hoc test named Nemenyi test has clearly tell us that which group individually different from other. This test is performed manually. In Table 53 the significant levels (p-value) for each algorithm is arranged in ascending order. The dataset has seven (k = 7) different algorithms' data and each algorithm is run for 50 times (N). By using the relation $Q_{12} = \frac{|R1-R2|}{\sqrt{\frac{k(k+1)}{6N}}}$, the Q test statistic of Nemenyi test is calculated which are further used in the pairwise comparison. In this relation, k is the number of algorithms, N is number of runs, and $R_1$ as the average rank of respective algorithm. The procedure is shown as below in Table 56:

The values of Q are compared with the critical value of Q (Q-c), this Q-c is obtained from the critical value table of Nemenyi test, i.e., 2.949 when number of groups is 7 [42]. This comparison shows that there is no significant difference statistically among any pair. Because in case of Nemenyi test if the value of test statistic Q is less than the selected critical value then the null hypothesis (i.e., there is no significant difference between the performance of all groups) is retained. But if the decision is about the selection of best one to find the optimal solution then WOA (has lowest value of mean rank) is superior to all according to the result of Table 53.

## 6 Conclusion

This article has implemented different algorithms for finding the optimum value of the gains of the PI controller for the sensorless BLDC motor's speed with minimum steady-state error and keeping the system stable as much as possible. The transient analysis shows that the whale optimization algorithm performs well regarding minimum error and robustness. Meta-heuristic algorithms adapt random strategies to locate globally the optimum solution instead of following the gradient trajectory. Therefore, the depiction of the supremacy of one algorithm over another statistical analysis is necessary. Statistically, it is concluded that in regards to find the overall optimum value of the cost function, the WOA is the best.

According to the ANOVA test, these techniques are different from each other significantly. When the t-test is performed, it is concluded that there is a significant difference in the performance of WOA, APSO, LFWOA CMLFWOA, PSO, and ACO according to the results of significant value (2-tailed), which is less than 0.05; hence the null hypothesis (i.e., there performance of all algorithms is alike.) is violated. The mean rank comparison of algorithms, as shown in the Man Whitney U test, shows that WOA has less value of mean rank therefore,

the whale optimization algorithm performs better than others do. In other non-parametric tests like Wilcoxon signed-rank test, results show that there is significant difference between the performances of other algorithms and WOA except PSO_w. The large number of positive ranks also present the choice of WOA in solving this problem is best.

The Friedman test (with post hoc tests like Bonferroni-Dunn and Holm's Step-Down) results also indicates the significant difference among the performances of these algorithms while PSO_w has no statistical difference in performance as compared to WOA. The Friedman aligned ranks test is the extension of the conventional Friedman test, according to the results of this test and post hoc test (Nemenyi test) there is statistically no difference among the performance of these algorithms, but still WOA is the best choice according the mean rank value of aligned rank of WOA. Hence, if it is about the performance supremacy then WOA is superior to all these discussed techniques. The results of the Friedman aligned ranks presents statistically no significant difference while the results of most of performed tests reject the null hypothesis and show statistically a significant difference between the performance of these algorithms, except PSO_w.

## Author Contributions

**Conceptualization:** Fizza Shafique, Muhammad Salman Fakhar.

**Formal analysis:** Fizza Shafique, Muhammad Salman Fakhar.

**Investigation:** Fizza Shafique, Muhammad Salman Fakhar, Akhtar Rasool.

**Methodology:** Fizza Shafique, Muhammad Salman Fakhar.

**Project administration:** Muhammad Salman Fakhar.

**Resources:** Muhammad Salman Fakhar.

**Software:** Muhammad Salman Fakhar.

**Supervision:** Muhammad Salman Fakhar, Akhtar Rasool, Syed Abdul Rahman Kashif.

**Validation:** Fizza Shafique, Muhammad Salman Fakhar, Akhtar Rasool, Syed Abdul Rahman Kashif.

**Visualization:** Muhammad Salman Fakhar, Akhtar Rasool, Syed Abdul Rahman Kashif.

**Writing – original draft:** Fizza Shafique, Muhammad Salman Fakhar, Akhtar Rasool, Syed Abdul Rahman Kashif.

**Writing – review & editing:** Fizza Shafique, Muhammad Salman Fakhar, Akhtar Rasool, Syed Abdul Rahman Kashif.

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
