## [Decision Letter · Decision Letter 0]

9 Jun 2024

PONE-D-24-19658Analyzing the Performance of Metaheuristic Algorithms in Speed Control of Brushless DC Motor: Implementation and Statistical ComparisonPLOS ONE

Dear Dr. Rasool,

Thank you for submitting your manuscript to PLOS ONE. After careful consideration, we feel that it has merit but does not fully meet PLOS ONE’s publication criteria as it currently stands. Therefore, we invite you to submit a revised version of the manuscript that addresses the points raised during the review process.

**Please, refer to the reviewers comments for further details and revise your paper accordingly.**

We look forward to receiving your revised manuscript.

Kind regards,

Dr. Suhail

Academic Editor

PLOS ONE

Journal Requirements:

Reviewers' comments:

Reviewer's Responses to Questions

**Comments to the Author**

1. Is the manuscript technically sound, and do the data support the conclusions?

Reviewer #1: Yes

Reviewer #2: Yes

Reviewer #3: Partly

2. Has the statistical analysis been performed appropriately and rigorously? 

Reviewer #1: N/A

Reviewer #2: Yes

Reviewer #3: No

3. Have the authors made all data underlying the findings in their manuscript fully available?

Reviewer #1: Yes

Reviewer #2: Yes

Reviewer #3: No

4. Is the manuscript presented in an intelligible fashion and written in standard English?

Reviewer #1: Yes

Reviewer #2: Yes

Reviewer #3: No

5. Review Comments to the Author

**Reviewer #1:** This paper presented the Analyzing the Performance of Metaheuristic Algorithms in Speed Control of Brushless DC Motor: Implementation and Statistical Comparison. The paper is organized, and the authors describe objectives and methodology of their work. Suggest this paper could be reconsidered after revision. The following comments need to be incorporated in this revision.

1.Please ensure all abbreviations are explained the first time they are mentioned.

Abstract should be improved. Explain a problem and its proposed solution and benefits over existing strategies.

2.How to set optimal parameters in your model? You use standard heuristic model thus optimization of parameters is necessary to make it more efficient.

3.What are the needs and benefits of this paper? What are the challenges and contributions to this field? 4. Why a new method is needed? What are the limitations of existing ones? After a comprehensive discussion about the contributions and novelties, please, list them briefly. Please explain more clearly.

4.The background of the research is rather weak. The authors should provide a more thorough overview of the scientific problem and show unresolved issues.

5.What is the need of a new algorithm, when effective optimization algorithms are available in the literature?

6. Introduction is vaguely written. My suggestion is to divide the introduction into three subsections: 1) motivation and incitement, 2) literature review and 3) contribution and paper organization.

7.Improve the quality of figures.

8.There is no discussion on the cost effectiveness of the methods surveyed. What is the computational complexity? Please include such discussions.

9.There is no statistical test such as CEC 2019, 2017 and CEC 2022, to judge about the significance of the method's results. Without such a statistical test, the conclusion cannot be supported

**Reviewer #2:** 1. Introduction section should have a short summary at the end about the contents of the article for the readers. Kindly add.

2. Kindly present the previous works in a separate section, Literature review. This could be your section 2 as per traditional research methodology approach.

3. Section 3 PI controller is more of a background, so it can be added as a subsection named Background in section 2, literature review.

4. Please redraw the Fig 2. Flow chart of implementation of metaheuristic algorithm-PI control BLDC motor, using suitable tool, it looks pretty wayward. Also, please mention YES/NO on every decision box to make it clear for readers. Use of colors is highly recommended for this flow chart.

5. Kindly add fig. 2 in Methodology section instead of results section

6. Kindly add simulation model specification in section 5 results and discussion instead of Methodology section.

7. It is strongly suggested to concise the section 5 in graphical form as it is convenient form for the readers to understand. The results are heavily populated with so many tables, try and replace as many with graphs.

8. Please include some relevant citations from year 2024. Also, it is recommended to cite more recent works from 2023 and 2022, since majority of references included are old.

**Reviewer #3: **This paper presents the implementation and statistical analysis of metaheuristic algorithms in speed control of brushless DC motor. However, there are some issues that need to be resolved. See the attachment.

6. PLOS authors have the option to publish the peer review history of their article (what does this mean?). If published, this will include your full peer review and any attached files.

Reviewer #1: No

Reviewer #2: No

Reviewer #3: No

---

## [Author Response · Author response to Decision Letter 0]

9 Jul 2024

The authors are thankful to the editor and the reviewers for their valuable time in providing their important and critical comments to improve the manuscript. We have provided the point-by-point responses in the attached file named "Response to Reviewer Comments" as per the guidelines. It was not possible to add equations, tables, etc. in this box so lease refer to the"Response to Reviewer Comments" attachment. Thanks

---

## [Decision Letter · Decision Letter 1]

29 Jul 2024

PONE-D-24-19658R1Analyzing the Performance of Metaheuristic Algorithms in Speed Control of Brushless DC Motor: Implementation and Statistical ComparisonPLOS ONE

Dear Dr. Rasool,

Thank you for submitting your manuscript to PLOS ONE. After careful consideration, we feel that it has merit but does not fully meet PLOS ONE’s publication criteria as it currently stands. Therefore, we invite you to submit a revised version of the manuscript that addresses the points raised during the review process.

We look forward to receiving your revised manuscript.

Kind regards,

Muhammad Suhail, Ph.D

Academic Editor

PLOS ONE

Journal Requirements:

Reviewers' comments:

Reviewer's Responses to Questions

**Comments to the Author**

1. If the authors have adequately addressed your comments raised in a previous round of review and you feel that this manuscript is now acceptable for publication, you may indicate that here to bypass the “Comments to the Author” section, enter your conflict of interest statement in the “Confidential to Editor” section, and submit your "Accept" recommendation.

Reviewer #1: (No Response)

Reviewer #2: All comments have been addressed

2. Is the manuscript technically sound, and do the data support the conclusions?

Reviewer #1: Yes

Reviewer #2: Yes

3. Has the statistical analysis been performed appropriately and rigorously? 

Reviewer #1: N/A

Reviewer #2: Yes

4. Have the authors made all data underlying the findings in their manuscript fully available?

Reviewer #1: (No Response)

Reviewer #2: Yes

5. Is the manuscript presented in an intelligible fashion and written in standard English?

Reviewer #1: (No Response)

Reviewer #2: Yes

6. Review Comments to the Author

Reviewer #1: The changes should be highlighted. I can see the responses to my comments except few， but I cannot see the changes in the revised manuscript.

1.How to set optimal parameters in your model? You use standard heuristic model thus optimization of parameters is necessary to make it more efficient.

2.There is no statistical test such as CEC 2019, 2017 and CEC 2022, to judge about the significance of the method's results. Without such a statistical test, the conclusion cannot be supported

3.What is innovative about the method proposed in this article? Need to explain clearly

Reviewer #2: The authors have compiled all the suggestions made and paper looks in good to be published. The authors efforts are appreciated.

7. PLOS authors have the option to publish the peer review history of their article (what does this mean?). If published, this will include your full peer review and any attached files.

Reviewer #1: No

Reviewer #2: No

---

## [Author Response · Author response to Decision Letter 1]

15 Aug 2024

Author Response

The authors are thankful to the editor and the reviewer for providing their valuable time and valuable comments to improve our article. Below, please find the point-wise response including relevant improvements and as well as addition of new results. The main manuscript has also been updated comprehensively with additional results for the verification and validation of the algorithm. 

Point-wise Response to Reviewer’s Comments

1. How to set optimal parameters in your model? You use standard heuristic model thus optimization of parameters is necessary to make it more efficient.

Response

Thank you for sharing your valuable thoughts. We tried our best to address your concerns with a detailed response. 

The main concern of this study is not related to the tuning of parameters values or range of values (coefficients of exploration and exploitation) of metaheuristic algorithms’ structure. We are interested in observing the outcomes of the predefined parameters of the relative algorithm with regard to our specific problem, i.e., gauging the gains of a PI controller for speed control of the sensorless BLDC motor. Different models like linearly decaying model is utilized to vary the values of a1 and a2 in PSO and APSO models, and the coefficients of exploration and exploitation in model WOA technique is varied by incorporated Levy flight trajectory and further modified using chaotic map. 

The target is to establish a comparative environment to check the efficiency of different metaheuristic techniques to find the optimal one for the concerned problem. This finding does not claim this is ultimately the best technique for other problems too. According to the “No Free Lunch Theorem”, the best performance algorithm for one problem never guarantee the best solution for other problems. We can make conclusion on the basis of this theorem that WOA gives best optimal solution for this specific model of motor and working conditions. It is also obvious that we have to make one condition persistent and vary the other, it means to compare the performances of different algorithms is differ than the tuning of relevant parameters of the models of these metaheuristic techniques. To tune the parameters of a certain algorithm may be implemented by us or any other researcher in this field, but for this purpose we have to keep the solution value as fixed value and set the range of values of exploration and exploitation controlling coefficients relative to the required solution. In a nutshell, if we say this study is about the comparison of different techniques and their already provided improved versions to tune PI controller’s gain. 

2. There is no statistical test such as CEC 2019, 2017 and CEC 2022, to judge about the significance of the method's results. Without such a statistical test, the conclusion cannot be supported.

Response

We highly regard your comments and appreciate your concern your suggestion of implement strong statistical tests which are discussed in CEC 2017, 2019, and 2022, for inclusion. 

In this paper, we have already performed rigorous statistical tests such as the Mann Whitney U test, the T test, and One-way ANOVA, with some concerned pos hoc tests including the Bonferroni test, the Tukey HSD test, and the Scheffe test. But we also understand your concern; therefore, we have attended to this remark and studied the recommended specific statistical tests or techniques that are generally applied in these competitions. Moreover, we tried to conclude the results with strong foundations more clearly.

In response, we have examined the specific statistical methodologies recommended and commonly used in CEC 2017, 2019, and 2022 competitions. We have supplemented our analysis with the recommended tests (e.g., Friedman test and relevant post-hoc analysis) and included a detailed comparison of these methods with our previously used tests (Mann-Whitney U Test (also known as Wilcoxon rank-sum test), One-Way ANOVA, and T-Test). This additional analysis has been incorporated into the revised manuscript. We believe this enhances the robustness of our conclusions and aligns our study with established standards in the field of evolutionary computation. In below description we have highlighted modified text from the revised manuscript.

The modifications are done in Table 35-46, and 51-56, and references 30-40, and 41. The modified text and results are highlighted in the attached modified manuscript. 

(Modified text in Contribution Section)

Furthermore, this article creates a comprehensive framework for the evaluation of the performance of these algorithms. The analytical comparison is done on the basis of the value of cost function that is minimization of error based on ISE criteria. To check the statistical significance, hypothetical testing is performed using statistical tests like ANOVA (Analysis of Variance), Mann Whitney U test, t-test, Wilcoxon signed-rank test, Friedman test, and Friedman aligned ranks test in SPSS software.

(Modified text in Section of Paper Organization)

Section 5 compromises of the simulated results and tabular data describes 117 results of different statistical tests (T test, Mann Whitney U test, Wilcoxon signed-rank 118 test, Friedman test, One-way ANOVA test), and Friedman aligned ranks test (improved 119 version of Friedman test).

4.8.1 T-test: 

T-test is a widely used statistical tests which is used to evaluate the significant difference of two groups on the basis of their mean. In early 1900s an English chemist and statistician named William Sealy Gosset published a seminal paper on t-test using the pseudonym “Student”. Therefore, the other name of t-test is Student’s t-test [33]. T-test is classified as paired samples t-test, independent samples t-test, and one sample t-test. The paired samples t-test is used for the comparison of same data for two different conditions, and the one sample t-test is used to compare a single group against any reference value. The independent samples t-test as name suggests is used to find statistically a significant difference between two non-related groups on the basis of their mean [34]. In this study, the independent samples t-test is applied to compare the means of algorithms in different sets.

4.8.2 Mann-Whitney U test:

In 1947, professor Henry Mann and his fellow D. Ransom Whitney published their work under the name of “On a test of whether one of two random variables is stochastically greater than the other”. In that seminal paper, they proposed a new statistical test named Mann-Whitney U test [35]. This test is basically used for the data set for which there is no need to assume normality in distribution. Mann-Whitney U test finds the significant difference between two groups by comparing their respective mean ranks instead of means.

4.8.3 Wilcoxon signed-rank test: 

The Wilcoxon signed-rank test was developed in 1945 by Frank Wilcoxon [36]. This is another non-parametric test which is used to determine the statistical difference either exist or not between two groups. In this set a positive or negative sign is assigned to each rank. This test can be used for small sized samples.

4.8.4 One-way analysis of variance (ANOVA):

 In 1920s, Sir Ronald A. Fisher introduced a new statistical approach that analyzes the variance among more than one group at a same time [37]. It is a parametric test that is helpful to compare large number of groups. One-way ANOVA provides the information of variance between different groups and variance within each group. One-way ANOVA provides the overall comparison of each group while to separately identify different pairs’ comparison Post Hoc tests become necessary. In this study, some these tests are performed like Bonferroni, Scheffe, and Tukey HSD (Honestly Significant Difference) tests.

4.8.5 Friedman test: 

The Friedman test detects the significant difference between more than two groups simultaneously like one-way ANOVA. But the Friedman test is a non-parametric test; therefore, there is no need to hold the assumption of uniformly distributed data. The Friedman test was first introduced by an American statistician, Milton Friedman [38]. The Friedman test is performed on data set which is organized in the form of blocks. The data in each block is ranked first then the ranks across each block are summed. In this test the significant difference level is determine by the comparison of test statistics with chi-square distribution [39]. Due to multiple comparison the risk of type 1 error increases; therefore, post hoc tests (Bonferroni-Dunn, and Holm’s Step-Down) are performed manually to provide the information about comparison of individual group.

4.8.6 Friedman aligned ranks test: 

The Friedman aligned ranks test is the modified variant of the conventional Friedman test. In this test, the block effect is neutralized. This effect of block is neutralized by aligning the data and assigning the rank to each group. Next step is the adjustment of data by subtracting the individual rank of each block from the mean rank of whole data. In this test, statistically a significant difference is found by comparing the aligned ranks with the value of chi-square [40]. Like other multi-comparison tests, there are some post hoc tests for the Friedman aligned ranks test. The most commonly used post hoc test is Nemenyi test.

5.8.3 Results for Wilcoxon signed-rank test: 

Table 35 to 36 provide the results of Wilcoxon signed-rank test of other algorithms vs. WOA. In Table 36 the negative Z-value suggests the performance supremacy of WOA over APSO. The Wilcoxon signed-rank test is performed for APSO-WOA; therefore, the value of Z has negative sign. Moreover, the associated p-value (Asymp. Sig. (2-tailed)) is also less than 0.05 which represents a significant statistical difference between APSO and WOA in error minimization. In Table 35 the large number of positive ranks also depicts that WOA is efficient than APSO.

The test statistics for ACO-WOA in Wilcoxon signed-rank test are shown in Table 38 in which the negative Z-value suggests the performance supremacy of WOA. In Table 38 the p-value (Asymp. Sig. (2-tailed)) is less than 0.05 and indicates a significant statistical difference between the performances of ACO and WOA. In Table 37 the number of positive ranks is larger than the negative ranks; therefore, the efficiency of WOA is higher than ACO.

Table 40 presents the test statistics of Wilcoxon signed-rank test for PSO-WOA, the negative Z-value suggests the performance supremacy of WOA and the p-value (Asymp. Sig. (2-tailed)), i.e., 0.000129 < 0.05 which indicates a statistical difference between the performances of PSO and WOA. The larger value of positive ranks number in Table 39 shows that WOA performs better than PSO.

Table 41 presents mean rank and sum of rank of the negative ranks and the positive ranks for Wilcoxon signed-rank Test of PSO-w-WOA. Number of positive ranks is larger than the negative ranks; therefore, WOA is efficient than PSO-w. In Table 42 the test statistics (Z-value and Asymp. Sig. (2-tailed), i.e., p-value (0.133) which is > 0.05) of Wilcoxon signed-rank test suggests the performance supremacy of WOA over PSO-w but there is statistically no significant difference between their performance.

In Table 43 the larger number of positive ranks presents the supremacy of WOA in finding the optimal solution. In Table 44 the test statistics of Wilcoxon signed-rank test for CMLFWOA-WOA, the negative Z-value presents the performance supremacy of WOA and the p-value (Asymp. Sig. (2-tailed)), i.e., 1.7569 × 10−9 < 0.05 which statistically shows a significant difference between the performances of both algorithms. 

Table 45 presents the mean rank and sum of ranks of positive ranks and negative ranks. The large number of positive ranks shows that WOA performs better than LFWOA. Table 46 presents the test statistics of Wilcoxon signed-rank test for LFWOA-WOA. The negative Z-value presents the performance supremacy of WOA and the p-value (Asymp. Sig. (2-tailed)), i.e., 0.000047 < 0.05 which statistically shows a significant difference 639 between the performances of both algorithms.

5.8.5 Results of Friedman test: 

The result of the Friedman test is show in Table 51. In this table the mean rank of each algorithm is provided. The value of mean rank for WOA is lowest which indicates that WOA is the most efficient algorithm than other algorithms. Table 52 presents the test statistics of the Friedman test, N is the size of data set which is ultimately the number of times each algorithm is run. In Table 52 the high value of chi-square, i.e., 133.037 shows significant difference between the performances of algorithms. The value of df (degree of freedom) presents the number of comparisons those are made independently. The Asymp. Sig. < 0.001 rejects the null hypothesis and shows a significant difference statistically between the performance of algorithms. In case of the Friedman test similar to the ANOVA test, due to multiple comparison there are chances of the occurrence of the type 1 error. Therefore, post hoc tests are applied to avoid the possibility of these errors. Holm’s Step-Down and Bonferroni-Dunn corrections are applied which promote the accuracy in results, and tell which algorithm (group) is statistically different from other. After Table 52 these corrections are applied.

The selected significance level α is 0.05 and total number of comparisons (k) is 6. Following are the steps for Holm’s adjustment and Bonferroni correction to show the individual difference of each algorithm from WOA and avoid the possibility of type 1 error. 

Holm’s Step-Down (Post Hoc test): 

The criteria which is followed in Holm’s adjustment procedure is α k+1−i (here i is the rank number of arranged p-value (presented in Wilcoxon signed-rank test statistics) of each comparison in ascending order as shown in Table 53. Using this relation of the Holm’s adjustment the following level of significance value is set for each comparison. 

1. Holm’s adjustment for ACO-WOA (smallest p-value) is 0.05/6 =0.008333. The p-value 680 for ACO-WOA in Table 53 i.e., 1.4686 × 10−9 which is less than 0.008333 that statistically indicates a significant difference between the performance of ACO and 682 WOA. 

2. Holm’s adjustment for CMLFWOA-WOA is 0.05/5 = 0.01. In Table 53 p-value for 684 CMLFWOA-WOA is 1.7569 × 10−9 which is less than 0.01. Therefore, statistically there is a significant difference between the performance of CMLFWOA and WOA. 

3. Holm’s adjustment for the comparison, LFWOA-WOA is 0.05/4 =0.0125. From Table 53 the p-value corresponds to the LFWOA-WOA comparison is 0.000047, i.e., < 0.0125 and presents a significant difference between the performance of LFWOA and WOA. 

4. Holm’s adjustment for PSO-WOA is 0.05/3 =0.016667. Comparison of the p-value 692 for PSO-WOA (from Table 53) i.e., 0.000129 which is less than 0.016667. It shows statistically a significant difference between the performance of PSO and WOA. 

5. Holm’s adjustment for APSO-WOA is 0.05/2 =0.025. The p-value of APSO-WOA comparison in Table 53 is 0.000214 which is less than 0.025 that statistically indicates a significant difference between the performance of APSO and WOA. 6. Holm’s adjustment for PSO-w-WOA (largest p-value) is 0.05/1 =0.05. The p-value for the comparison of PSO-w-WOA in Table 53 i.e., 0.133334 which is greater than 0.05. This p-value indicates that there is statistically no significant difference between the performance of PSO-w and WOA.

Bonferroni-Dunn (Post Hoc test): 

This correction is helpful in avoiding the possibility of type 1 error due to multiple comparison on same data set. The decision rule for selecting the significance level for each comparison is done by using a new significance threshold, i.e., α/k. In this case the new adjusted threshold is 0.008333, now the decision about each comparison depends on this value. By comparing each p-value in Table 53, it is found that there is a significant difference between ACO and WOA, CMLFWOA and WOA, LFWOA and WOA, PSO and WOA, and APSO and WOA except PSO-w and WOA. Because 0.133334 > 0.008333 which statistically shows no difference among the performances of PSO-w and WOA techniques.

5.8.6 Res

---

## [Decision Letter · Decision Letter 2]

26 Aug 2024

Analyzing the Performance of Metaheuristic Algorithms in Speed Control of Brushless DC Motor: Implementation and Statistical Comparison

PONE-D-24-19658R2

Dear Dr. Akhtar Rasool,

We’re pleased to inform you that your manuscript has been judged scientifically suitable for publication and will be formally accepted for publication once it meets all outstanding technical requirements.

Kind regards,

Muhammad Suhail, Ph.D

Academic Editor

PLOS ONE

Additional Editor Comments (optional):

Reviewers' comments:

Reviewer's Responses to Questions

**Comments to the Author**

1. If the authors have adequately addressed your comments raised in a previous round of review and you feel that this manuscript is now acceptable for publication, you may indicate that here to bypass the “Comments to the Author” section, enter your conflict of interest statement in the “Confidential to Editor” section, and submit your "Accept" recommendation.

Reviewer #1: All comments have been addressed

2. Is the manuscript technically sound, and do the data support the conclusions?

Reviewer #1: Yes

3. Has the statistical analysis been performed appropriately and rigorously? 

Reviewer #1: Yes

4. Have the authors made all data underlying the findings in their manuscript fully available?

Reviewer #1: Yes

5. Is the manuscript presented in an intelligible fashion and written in standard English?

Reviewer #1: Yes

6. Review Comments to the Author

Reviewer #1: (No Response)

7. PLOS authors have the option to publish the peer review history of their article (what does this mean?). If published, this will include your full peer review and any attached files.

Reviewer #1: No

---

## [Editor Report · Acceptance letter]

2 Sep 2024

PONE-D-24-19658R2 

PLOS ONE

Dear Dr. Rasool, 

I'm pleased to inform you that your manuscript has been deemed suitable for publication in PLOS ONE. Congratulations! Your manuscript is now being handed over to our production team.

Kind regards, 

on behalf of

Dr. Muhammad Suhail 

Academic Editor

PLOS ONE